**Development and validation of the Terrain Stability model for assessing landslide instability during heavy rain infiltration.**

1. Alfonso Gutiérrez-Martín[1] Miguel Ángel Herrada[2], José Ignacio Yenes Gallego[3], Ricardo Castedo Ruiz[4]

[1] Dr. Arquitecto, Escuela Superior de Arquitectura; Universidad de Málaga, España. E-mail: alfgutmar@uma.es

[2] Catedrático de Universidad, Escuela Superior de Ingenieros de la Universidad de Sevilla. España. E-mail: herrada@us.es

[3] Dr. Ingeniero José Ignacio Yenes Gallego, Jefe de Unidad, Dirección General de Infraestructuras, MINISDEF, Madrid, España. E-mail: jyengal@et.mde.es

[4] Dr. Ingeniero Ricardo Castedo Ruiz, Departamento de Ingeniería Geológica y Minera, Universidad Politécnica de Madrid, España. E-mail: ricardo.castedo@upm.es

## Abstract

Slope stability is a key topic, not only for engineers but also for politicians, due to the considerable monetary and human losses that landslides can cause every year. In fact, it is estimated that landslides have caused thousands of deaths and economic losses amounting to tens of billions of e6uros per year around the world. The geological stability of slopes is affected by several factors, such as climate, earthquakes, lithology and rock structures, among others. Climate is one of the main factors, especially when large amounts of rainwater are absorbed in short periods of time. Taking into account this issue, we developed an innovative analytical model using the limit equilibrium method supported by a geographic information system (GIS). This model is especially useful for predicting the risk of landslides in scenarios of heavy unpredictable rainfall. The model, hereafter named 'Terrain Stability' or TS is a 2D model, programmed in MATLAB and includes a steady state hydrological term. Many variables measured in the field – topography, precipitation, type of soil – can be added, changed or updated using simple input parameters. To validate the model, we applied it to a real example, that of a landslide which resulted in human and material losses (collapse of a building) at Hundidero, La Viñuela (Málaga), Spain, in February 2010.

**Keywords:** Rainfall, Slope, Limit equilibrium model, algorithm and critical surface.

## 1. Introduction

Landslides, one of the natural disasters, have resulted into significant injury and loss to human life and damaged property and infrastructure throughout the world (Varnes, 1996; Parise and Jibson, 2000; Dai et al., 2002; Guha-Sapir et al., 2004; Crozier and Glade, 2005; Kahn, 2005; Toya and Skidmore, 2007; Raghuvanshi et al., 2014; Girma et al., 2015). Normally, heavy rainfall, high relative relief and complex fragile geology with increased manmade activities, have resulted in increased landslide (Gutiérrez-Martín, 2015). It is essential to identify, evaluate and delineate landslide hazard prone areas for proper strategic planning and mitigation (Bisson et al., 2014). Therefore, to delineate landslide susceptible slopes over large

Gutiérrez-Martín, A.

areas, landslide hazard zonation (LHZ) techniques can be employed (Anbalagan, 1992;
Guzzetti et al., 1999; Casagli et al., 2004; Fall et al., 2006).
Landslides are resulted because of intrinsic and external triggering factors. The intrinsic
factors are mainly; geological factors, geometry of the slope (Hoek and Bray, 1981; Ayalew et
al., 2004; Wang and Niu, 2009).
The external factors which generally trigger landslides are rainfall (Anderson, 1985; Collison et
al., 2000; Dai and Lee, 2001). Several LHZ techniques have been developed over the past and
these can be broadly classified into three categories; expert evaluation, statistical methods
and deterministic approaches (Wu and Sidle, 1995; Leroi, 1997; Guzzetti et al., 1999; Inverson,
2000; Crosta and Frattini, 2003; Casagli et al., 2004; Fall et al., 2006; Lu and Godt, 2008; Rossi
et al., 2013; Raia et al., 2014; Canili et al., 2018; Zhang et al.; 2018). Within these categories,
we want to highlight the empirical models that are based on rainfall thresholds (Wilson, 1997;
Aleotti, 2004; Gruzzetti et al., 2007; Martelloni et al., 2011). Each of these LHZ techniques has
its own advantage and disadvantage owing to certain uncertainties on account of factors
considered or methods by which factor data are derived (Carrara et al., 1995). Limit
equilibrium types of analyses for assessing the stability of earth slopes have been in use in
geotechnical engineering for many decades. The idea of discretizing a potential sliding mass
into vertical slices was introduced in the 20th century. During the following few decades,
Fellenius (1936) introduced the Ordinary method of slices (Fellenius, 1936). In the mid1950s
Janbu and Bishop developed advances in the method (Janbu, 1954; Bishop, 1955). The advent
of electronic computers in the 1960's made it possible to more readily handle the iterative
procedures inherent in the method, which led to mathematically more rigorous formulations
such as those developed by Morgenstern and Price and by Spencer (Morgenstern and Price,
1965; Spencer, 1967).
Until the 1980s, most stability analyses were performed by graphical methods or by using
manual calculators. Nowadays, the quickest and most detailed analyses can be performed
using any ordinary computer (Wilkinson et al., 2002). There are other types of software based
on the modeling of the probability of occurrence of shallow landslides LHZ, in more extensive
areas using GIS technology and 'DEM' (Digital Elevation Model), as is the case of deterministic
models like: software TRIGRS, SINMAP, R-SHALSTAB, GEOtop/GEO-FS, R-Slope-stability among
others (Montgomery and Dietrich, 1998; Pack et al., 2001; Rigon et al., 2006; Simoni et al.,
2008 ; Baum et al., 2008; Mergili et al., 2014a; Mergili et al., 2014b; Michel et al., 2014; Reid et
al., 2015; Alvioli and Baum, 2016; Tran et al., 2018). These are widely used models for
calculating the time and location of the occurrence of shallow landslides caused by rainfall at
the territorial level; some even in three dimensions, in order to obtain a probabilistic
interpretation of the factor of safety. Currently other approaches / theoretical studies for
landslide prediction are used (for triggering and / or propagation) (Martelloni and Bagnoli,
2014; Martelloni et al., 2017). One of the achievements of the presented study is to discretize
the potential slip mass in the critical profile of the slope, once unstable areas have been
detected through the 'LHZ' (landslide hazard zonation) programs. The TS calculation tool is not
limited to shallow landslides and debris flows, but allows analysis of deep and rotational
landslides, which often other models do not accommodate for. We use in our algorithm the

Gutiérrez-Martín, A.

hydrological variable '$r_u$' of Spencer, to consider the infiltration of rainfall in the calculation of
stability of the considered slope.
Limit equilibrium types of analyses for assessing the stability of earth slopes have been in use
in geotechnical engineering for last years. Currently, the vast majority of stability analyses
using **this method of equilibrium limit** are performed with commercial software packages like
SLIDE V5, SLOPE/W, Phase2, GEO-Slope, GALENA, GSTABL7, GEO5  and GeoStudio, among
others (Gonzalez de Vallejo et al., 2002; Acharya et al., 2016a; Acharya et al., 2016b; Johari and
Mousavi, 2018). Currently there are other slope stability models based on the theory of limit
equilibrium that are still in analysis and testing, as is the case with the SSAP software package
(Borselli, 2012), but in this case a general equilibrium method model is applied. Secondly,
sometimes for commercial models, the introductions of parameters to perform calculations
are not very interactive. For the stability analysis, different approaches can be used, such as
the limit equilibrium methods (Cheng et al., 2007; Liu et al., 2015), the finite elements method
(Griffiths et al., 2007; Tschuchnigg et al., 2015; Griffiths, 2015) and the dynamic method (Jia et
al., 2008), among others. Limit equilibrium methods are well known, and their use is simple
and quick. These methods allow us to analyse almost all types of landslides, such us
translational, rotational, topple, creep and fall, among others (Zhou and Cheng, 2013). For the
stability analysis, different approaches can be used, such as the limit equilibrium methods (Zhu
et al., 2005; Cheng et al., 2007; Verruijt, 2010; Liu et al., 2015), the finite elements method
(Griffiths et al., 2007; Tschuchnigg et al., 2015; Griffiths, 2015) and the dynamic method (Jia et
al., 2008) among others (SSAP 2012, Slide V5-2018). Also, limit equilibrium methods can be
combined with probabilistic techniques [Stead et al., 2000] or with other models, like stability
analysis of coastal erosion (Castedo et al., 2012). However, they are limited in general to 2D
planes and easy geometries. Numerical methods – finite elements methods – give us the most
detailed approach to analysing the stability conditions for the majority of evaluation cases,
including complex geometries and 3D cases. Nevertheless, they present some problems, such
as their complexity, data introduction, mesh size effect and the time and resources they
require (Ramos Vásquez, 2017).
The above-mentioned software packages provide useful tools for determining the stability
through the $F_S$ (safety of factor) and for giving the most probable breakage (shearing) surfaces.
This technique is fast and allows the field or emergency engineer to make timely decisions.
Although this methodology is only available in some current software (Slide V 5.0, STB 2010,
Geo-Slope), and based on limit equilibrium methods, it is highly recommended because of its
reliability for representing real conditions in the field (Chugh, 1981). This rain infiltration
produces a substantial reduction of cohesion (a key soil parameter for stability) that cannot be
reproduced by actual software and then several real situations cannot be predicted.
Delft University has developed a well-known and free software programme to analyse
landslides, the STB 2010 (Verruijt, 2010). This programme is based on a limit equilibrium
technique, using a modified version of Bishop's method to calculate the $F_S$ only for circular
failures. It is a user-friendly tool, but it does not allow the calculation of water infiltration on a
hillside. This is a critical point, as it is well known that rainfall infiltration is one of the main
causes of landslides worldwide (Michel et al., 2015). Reviewing these issues, a new solution
must be developed for cases where landslides are linked to heavy rainfall. In this study, we

Gutiérrez-Martín, A.

developed a new model and programmed it using MATLAB. The primary result of this model
was a stability index, namely the minimum *Fs,* based on the limit equilibrium technique, in this
case the Bishop's method.  The model also provides a possible failure curve and surface area,
including the infiltration effects, which can be used to coincide with analysis of the actual
event as tested with field data. Topographical data can also be introduced into the model from
the digital elevation model (DEM) in a GIS.

## 2. Terrain Stability model development

In the model we developed the Terrain Stability (TS) model, we used the limit equilibrium
technique for its versatility, calculation speed and accuracy. An analysis can be done studying
the whole length of the breakage (shearing) zone or just small slices. Starting with the original
method of slides developed by Petterson and Fellenius (1936), some methods are more
accurate and complex (Spencer 1967; Morgenstern and Price, 1965) than others (Bishop, 1955
and Janbú, 1954). Using Spencer's method (Spencer, 1967; Chung, 1986) here would mean
dividing our slope into small slices that must be computed together. This method is divided
into two equations, one related to the balance of forces and the other to momentum.
Spencer's method imposes equilibrium not only for the forces but also for the momentum on
the surface of the rupture. If the forces for the entire soil mass are in equilibrium, the sum of
the forces between each slice must be also equal to zero. Therefore, the sum of the horizontal
forces between slices must be zero as well as the sum of the vertical ones (equations 1 and 2).

$$\sum[Q \cos \theta] = 0 \tag{1}$$

$$\sum[Q \sin \theta] = 0 \tag{2}$$

In this equation, $Q$ is the resultant of the pair of forces between slices, and $\theta$ is the angle of the
resultant (Figure 1). From this, it can be stated that the sum of the moments of the forces
between slices around the critical rotation centre is zero, conformed to equation 3:

$$\sum[QR \cos(\alpha - \theta) = 0] \tag{3}$$

When the R is the radius of the curvature, α is the angle of the slope referred to each slice. This
takes into account that the sliding surface is considered circular, so the radius of the curvature
is constant.

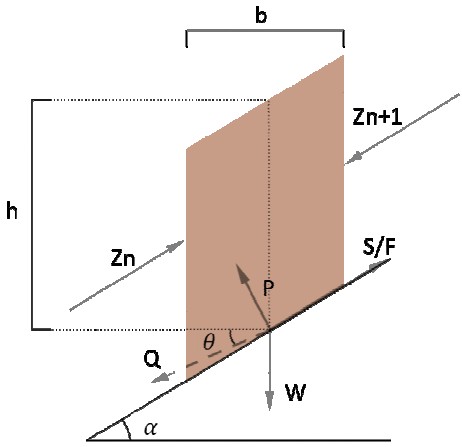


Gutiérrez-Martín, A.

*Figure 1. Representation of the forces acting on a slice, considered in Spencer's method (Spencer, 1967).*
*W is the external vertical loads; Zn and Zn+1 are the forces acting on the left- and right-hand side of each*
*slice, respectively, with their horizontal and vertical components; P and S are the normal and tangential*
*forces at the base of the slice; α is the angle of the slope referred to each slice, b is the slice width and h*
*is the mean height of slice (if the height is not constant).*
These equations must be solved to get the $F_s$, and tilt angles of the forces among the slices ($\theta$).
To solve these equations, an iterative method is required until a limiting error is reached. Once
$F_S$ and $\theta$ are calculated, the remaining forces are also obtained for each slice. Spencer's
method is considered very accurate and suitable for almost all kinds of slope geometries and
may be the most complete equilibrium procedure. It may also be the easiest method for
obtaining the $F_s$ (Duncan and Wright, 2005). Depending on the type of slope analysed, this
model is able to establish the failure curve following the typical rotational circle, among other
uses (Verruijt, 2010).
The $F_S$, classically defined as a ratio of stabilizing and destabilizing forces, determines the
stability of a slope as follows:
$$F_S = \frac{\sum(Forces\ standing\ against/oppose\ sliding)}{\sum(Forces\ that\ induce\ sliding)} \tag{4}$$

According to limit equilibrium methods, the two equilibrium conditions (forces and moments)
must be satisfied. Taking into account these elements, the Fs is then obtained from the
following expression (Spencer, 1967):
$$Fs = \frac{1}{\sum W \sin \alpha} \sum [c'b \sec \alpha + \tan \phi'\ (W \cos \alpha - ub \sec \alpha)] \tag{5}$$

Where $\phi'$ is the friction angle at the fracture surface, u is the pore pressure at the fracture
zone, c′ is the soil cohesion, $\alpha$ is the angle at the base of the slice, W is the external vertical
forces and b the width of the slice. According to equations (4) and (5), the slope can be
considered unstable if its value of the safety factor '$F_s$' is lower than 1, or stable if it is equal o
higher than 1. It should be noted that, when applying the factor in the engineering and
architecture fields, the limiting value tends to be higher than 1, with common values being 1.2
or even up to 1.5 (Burbano et al., 2009), security coefficients that include The European
technical regulations and, specifically, the technical regulations of Spanish application (table
2.1, of the DB-C of the CTE, or Technical Code of the Building) among others. This is just a
confidence measure for your calculations. The Fs can also be defined as the ratio between the
shear strength (τ), based on the cohesion and the angle of friction values, and the shear stress,
based on the cohesion and the internal friction angle required to maintain the equilibrium
($\tau_{mb}$).
As mentioned, the minimum Fs to consider a slope stable is equal to 1. However, several
authors (Yong et al., 1977; Van Westen and Terlien, 1996) suggest that the angle of a slope
would have to be defined by a value of the Fs superior to the unity to take into account the
exogenous factors of the slope. Following Jimenez Salas (1981), a value of $F_S \geq 1.3$ can be
considered stable by most standards.

Gutiérrez-Martín, A.

To analyse the slope using the Spencer's method, a set of equations must be solved to satisfy
the forces and momentum equilibrium and to obtain the $F_s$. The values of $F_s$ and $\theta$ are the
unknowns that must be solved. Some authors suggest that the variation of $\theta$ can be arbitrary
(Morgenstern y Price, 1965), although the effect of these variations in the final value of $F_s$ is
minimal. The variation of the angle depends on the soil's ability to withstand only a small
intensity of the shear stress.
Having said that, if we assume that the forces between slices are parallel (in other words, that
$\theta$ is constant), equations (1) and (2) become the same, resulting in:
$$\sum Q = 0 \tag{6}$$

The assumption that the forces between slices are parallel gives optimal results for the
calculation of the critical safety coefficients in equation 5 (Spencer, 1967). To solve these
equations, we used the FSOLVE function of the MATLAB software, giving an initial Fs and angle.
The FSOLVE function is a tool inside the optimization toolbox from MATLAB that solves
systems of nonlinear equations. When using this tool, an initial value must be provided to start
the calculation.
When solving the normal and parallel forces at the base of the slice of the five acting forces,
we obtain (Q), resulting from the forces between slices:
$$Q = \frac{\frac{c'b}{F}\sec\alpha + \frac{\tan\phi'}{F}(W\cos\alpha - ub\sec\alpha) - W\sin\alpha}{\cos(\alpha - \theta)[1 + \frac{\tan\phi'}{F}\tan(\alpha - \theta)]} \tag{7}$$

In this expression, u is the pore pressure (permanent interstitial pressure) at the base of the
slice and the weight of the slice is determined by W. If we assume that the soil is uniform and
its density (γ) also, the weight of a slice of height h and width b can be written:

$$W = \gamma b h \tag{8}$$

The application of a homogeneous pore pressure distribution (permanent interstitial pressure)
has been included in the model (Bishop and Morgenstern, 1960). In this case, the permanent
interstitial pressure on the base of the slice was determined by the following expression:
$$u = r_u \gamma h \tag{9}$$

In this expression, $u$ is the pore pressure (permanent interstitial pressure) at the base of the
slice, $\gamma$ is the density of soil, $h$ is the mean height of slice (if the height is not constant) and the
weight of it affects the $W$ evaluation.
The pore pressure will be hydrostatic, defined by: $u = \gamma_w(h - h_w)$, $\gamma_w$ is the saturated density
of soil, h and $h_w$ is the difference between saturated and dry height. The calculation of the
infiltration factor is calculated with the following equation:
$$r_u = \frac{u}{\gamma h} \tag{10}$$

The factor $r_u$ is a coefficient of pore pressure (interstitial pressure coefficient), which
determines the rain infiltration factor on the slopes. As it is well known, the water that

Gutiérrez-Martín, A.

infiltrates the soil may produce a modification of the pore pressure, affecting its resistant
capacity. This factor may vary from 0 (dry conditions) to 0.5 (saturated conditions). In the
article of Spencer (Spencer, 1967), assuming a homogeneous pore-pressure distribution as
proposed by Bishop and Morgenstern (1960), the mean pore-pressure on the base of the slice
can be written like the equation 7.
This equation is used in our proposed model for calculating the safety factor (substituting the
expression of *u* in equation 5).

## 3. Terrain stability (TS) algorithm and tests

Figure 2 shows the results of applying the Terrain Stability model to an irregular slope,
including the initial and final points of the first failure circle (shown in yellow). This circle
corresponds with the initial value introduced by the user into the FSOLVE function. The points
of the slope are extracted from a DEM model in ArcGIS 10 (Glennon et al., 2008). The slope
height is equal to 15 m, and the soil is considered uniform with the following nominal
properties: $\gamma$ = 19500 N/m$^3$, $\phi$ = 22º, $c$ = 15000 N/m$^2$, $u$ = 0 N/m$^2$. For the application example
of our algorithm in this section, we have used Geotechnical data of a cohesive soil of the Flysch
type of Gibraltar, (Vallejo et al., 2002).
The code works as follows: the initial circular failure curve is plotted using the FPLOT tool, as
shown in Figure 2 (yellow line). In this example, the center coordinates are equal to xc = 7 m;
yc = 14 m and the lower cut with the slope coordinates (P1 point) equal to xt = 0 m, yt = 0 m.
The Fs obtained was 1.6, which is, in principle, a stable slope. It must be taken into account
that the mass susceptible to sliding must be divided into a sufficient number of slices. This
value is entered into our code through the parameter 'N'. In the application example of our
algorithm, the sliding mass was divided into N = 500 slices, this value of N is entered into the
code by the user, who decides the value of that parameter. The greater the number of slices in
which we divide the sliding mass, the calculation will be more accuracy. N = 500 slice, we
consider it a balanced value for an optimal calculation, which relates two fundamental
parameters (computer calculation capacity / capacity accuracy).

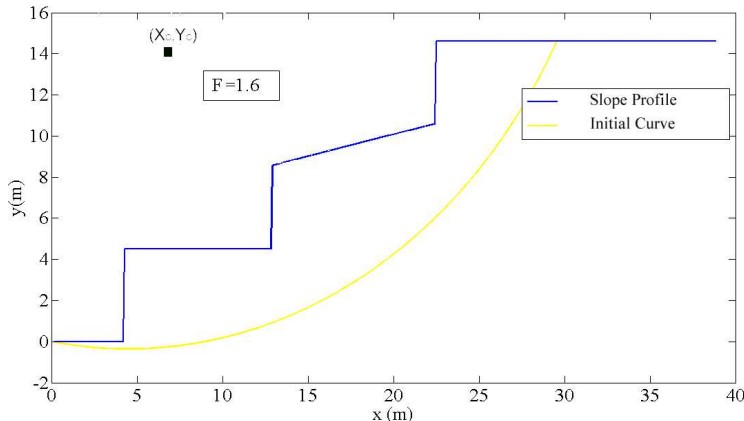


*Figure 2. Idealized cross section of a slope. In this example, the center coordinates are equal to xc = 7 m;*
*yc = 14 m, and the lower cut with the slope coordinates (P1 point) equal to xt = 0 m, and t = 0 m, data*
*that the user introduces.*

Gutiérrez-Martín, A.

The next step is to apply Spencer's method to the different breakage surfaces until the curve
with the lowest $F_S$, is found, and that will be the critical surface susceptible to a circular slip. To
determine the minimal Fs using this model, the algorithm calculates the displacement of the
lower cutoff point of the critical slip from the slope, as well as the position of the center of
rotation of the critical failure curve. In addition, the user must enter a series of possible
circular faults. Then, the user introduces the following constraints into the programme: the
initial or lower point of the failure curve ($P_1$) in its intersection point with the slope, which may
or may not match the origin of the slope analysed. Another restriction is the centre of the
failure circle, ($X_c$, $Y_c$), that should initially cut the slope, i.e. the breaking curve must be within
the feasible sliding region. With this data, the programme automatically draws a first curve, in
this case the yellow line in Figure 3, and calculates the safety coefficient $F_s$ for that initial curve.

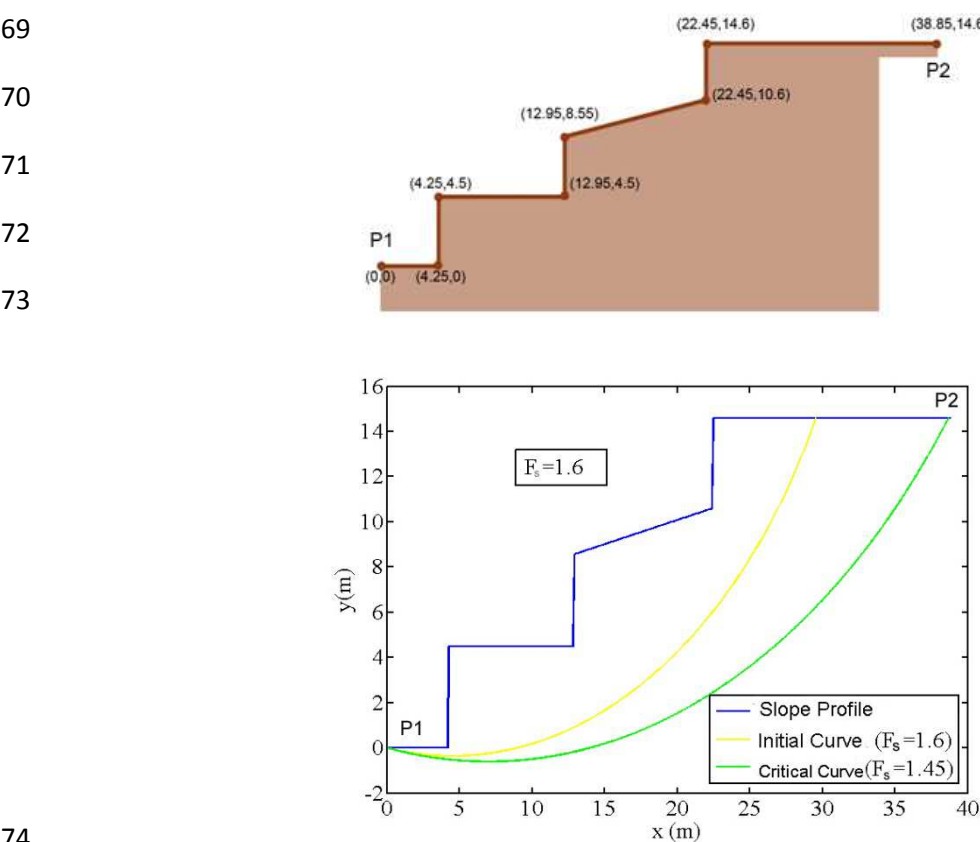






**Figure 3.** *Results following the application of the software showing the slope profile and surface*
*damage. The $F_S$ and the clearest proof of circular failure are also provided (see the yellow line). P1*
*coordinates are (0, 0) and P2 (38.85, 14.6) in metres.*
On the basis of this first curve (yellow line in Figure 2), the programme enforces new
restrictions:
• The curve passes through the origin of slope P1 = (0, 0).
• The centre of the possible circles of critical breakage is inside the rectangular box
defined as: ($x_{box\ min.} < x_c < x_{box\ max.}$; $y_{c\ box\ min.} < y_c < y_{box\ max}$). Note that the coordinates are
entered with the 2D expression (X, Y).

Gutiérrez-Martín, A.

Both coordinates of the rotation centre position are free and can change for every circle. From
the initial failure curve, characterised by the point x = ($x_c$, $y_c$), the MATLAB "fmincon" function
is used to obtain a new critical point ($x_c^*$, $y_c^*$) where the Fs from the breakage curve is the
minimum provided by fmincon. In this example, starting from the initial curve (yellow curve)
with point x = (7, 14), the TS model provides a new point x* = (4.4910, 28.1091, 0) with a new
Fs, $F_S$ = 1.45. In this case, the new search has been carried out with the following restrictions in
the rectangular box, such as 2 m < $x_c$ < 8 m and 16 m < $y_c$ < 40 m. These restrictions are
imposed in order to determine the critical circle. With all these restrictions, and because of the
first calculated curve (the yellow curve), the developed model calculates the critical curve
among the number of curves selected by the user (500 in this case), as well as the failure circle
centre, by applying the fmincon (MATLAB function). This defines the curve with minimum Fs
($F_{min}$) as the value of $F_s$ (see green curve in Figure 3). When solving this problem, a critical
selection is the lower cut-off point of the slope. According to different authors, such as Verruijt
(2010) and Castedo et al. (2012), the selected point is the same as the P1 point.
To complete the second phase in the TS model operation, the effect of rain infiltration must be
introduced by the coefficient of the pore pressure factor $r_u$. In this example, the infiltration
factor was introduced at the base of each slice to account for the infiltration and pore pressure
at the base of the break surface of the slope. If $r_u$ increases, the cohesion of the soil mass of
the slope decreases, directly affecting the reduction of the slope's Fs. The result is that a dry
slope has a $F_S$ = 1.45, but if including the $r_u$ parameter equal to 0.3, the Fs decreases to a value
of $F_S$ = 0.95, that means an Fs below the unity, so an unstable circular failure appears (see
Figure 4). Entering the infiltration factor, $r_u$, in Spencer's method to introduce the infiltration
effects in slopes, the geotechnical cutting elements of the analysed soil are reduced, also
reducing the values of the $F_s$, both for the initial yellow curve and the optimum green curve
(Figure 3). Note that the initial curve in the run shown in Figure 4 is different from the one in
Figure 3, as it depends on the data introduced.

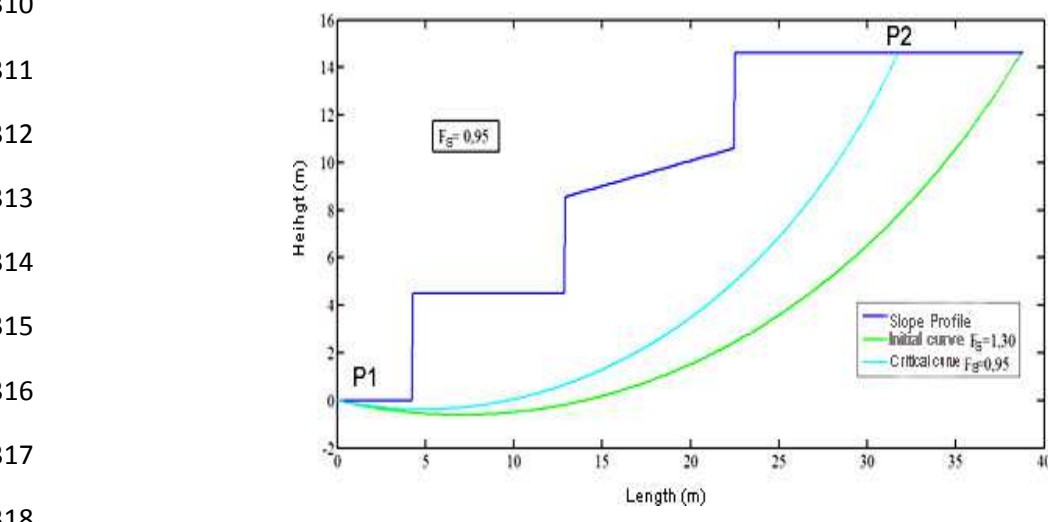

*Figure 4. Outcome of the TS model after the introduction of the infiltration factor, producing an unstable*
*circular failure (Fs = 0.95).*

Gutiérrez-Martín, A.

We can determine that if this infiltration factor value is small enough, taking into account the
safety coefficients, the design may still be adequate, but critical information was missing to
calculate this parameter.
To clarify the procedure employed in the suggested algorithm, the flowchart (block diagrams)
presented in Figure 5 demonstrates the calculation and iteration process as implemented in
our software.

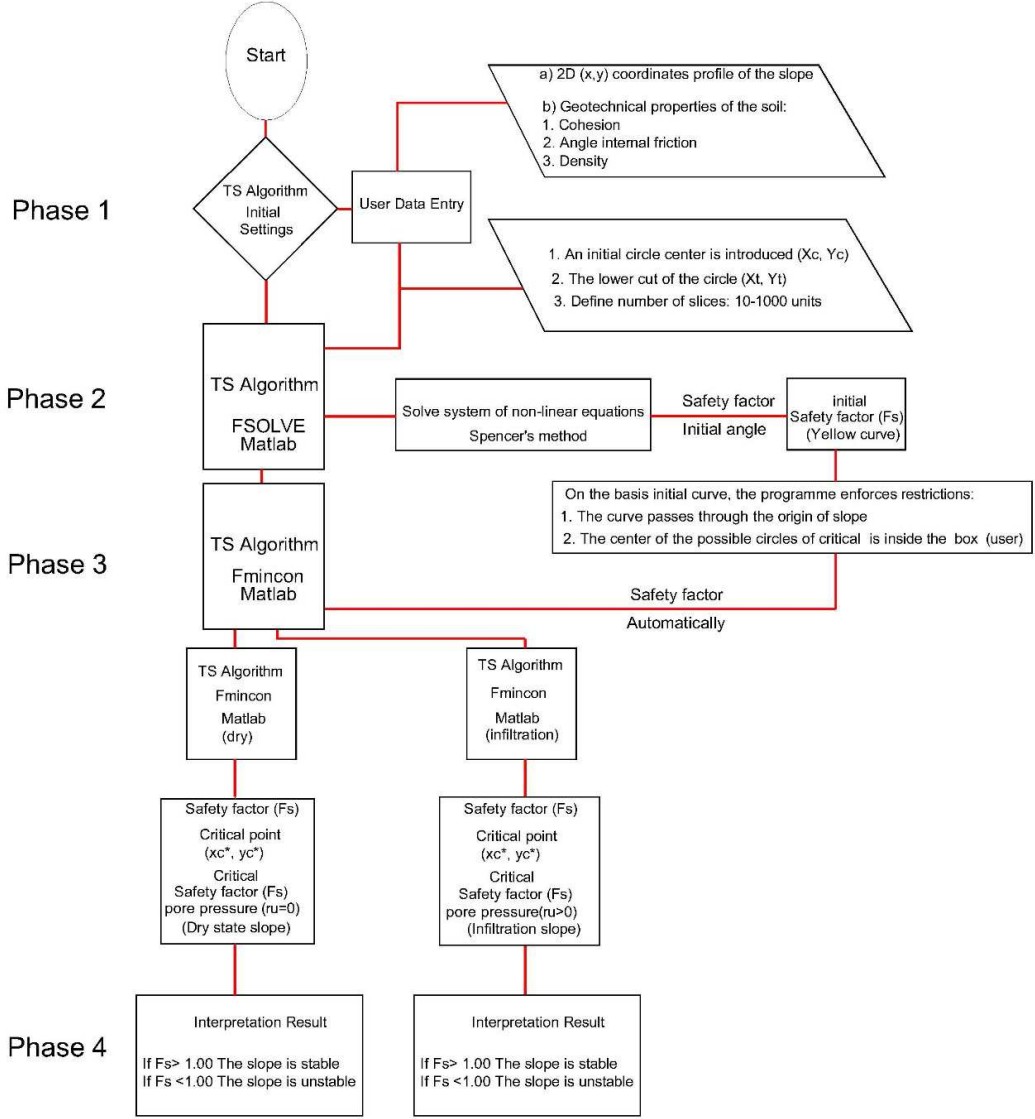

*Figure 5. Sequential TS algorithm (block diagrams). Numbers in parentheses refer to numbers in the*
*text.*
1. Our algorithm (software) is more versatile compared to the STB 2010, the model
developed here can analyze slope from right to left and vice versa, the STB 2010 only
allows the analysis from right to left. Other software programmes, like the STB 2010,
use a modified version of Bishop's method, a less accurate methodology than
Spencer's method. A modified version of Bishop's method solves only the equilibrium
in momentum while the Spencer method also considers the equilibrium in forces.

Gutiérrez-Martín, A.

2. Another improvement made by the TS code, in comparison with others, is that the use
of the Spencer's method allows us to analyse any type of slope and soil profile. In this
procedure, we calculated the worst breaking curve by modifying the calculation points.

3. In the TS model, from the first slip rotational circle obtained in MATLAB, many circles
were then calculated using the fmincon function, with some user restrictions.
However, other models, like the STB 2010, require the definition of a quadrangular
region (to look for the centres of rotational failures) and a point (namely 5, see Figure
9) to define the curve as where the failure must pass. Also, the number of circles that
the STB 2010 model can analyse for their minimum value is limited to 100.

4. The TS model can detect relevant earth movements derived from rainfall infiltration,
both translational and rotational types (Stead et al., 2006), such as those that usually
occur in regions like India, the United States, South America and the United Kingdom,
among other places. The programmes that do not contemplate this option will
overestimate the Fs, potentially with great errors.

The TS model has an additional advantage: it also offers the opportunity to incorporate, in the
same code, the stability analysis and the effect of the infiltration factor in the rainfall regime.
This is a step forward from open access programs, such as STB 2010, and also alternative
payment software, such as Slide.
**4. Example of this application in the municipality of La Viñuela, Málaga, Spain**
In 2010, La Viñuela, Málaga, (Spain) experienced torrential rainfall. The main consequence was
a devastating landslide with serious personal and material losses, as shown in Figure 6. The
coordinates where this event occurred were in degrees (36.88371409801, -4.204982221126).

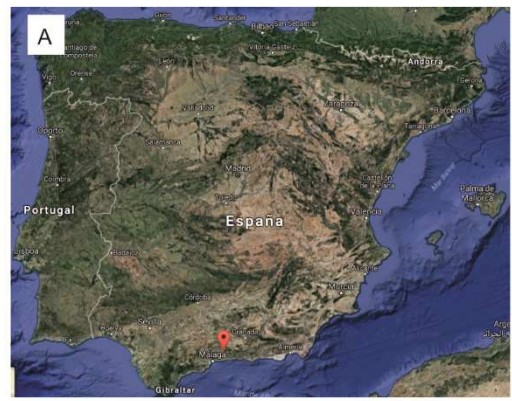

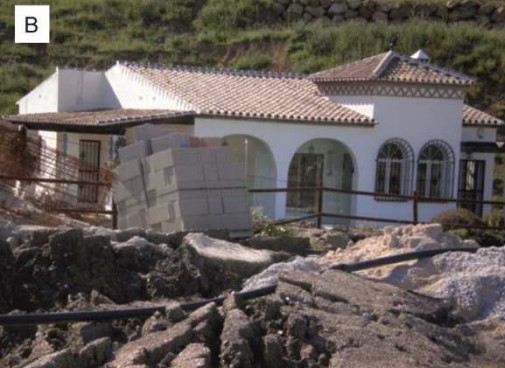 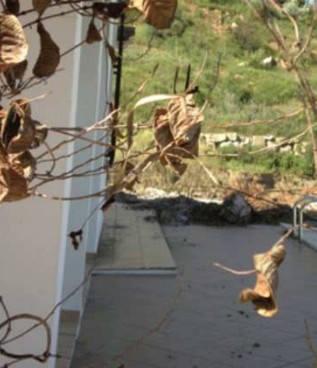


Gutiérrez-Martín, A.

**Figure 6.** *A) Spanish map with the location of La Viñuela (Google Maps). B) Real images taken by the authors at La Viñuela in 2010.*

### 4.1 Geological and hydrological environment

The study area is located in the county of La Viñuela, specifically in the Hundidero village, which is located immediately north of the swamp of La Viñuela (El Hundiero) and south of The Baetic System Mountain ranges (South Iberian Peninsula).

According to the Cruden and Varnes' classification (1996), the slide corresponds to a rotational slide-like complex movement because it was generated in two sequences at different speeds. This type of mechanism is characteristic of homogeneous cohesive soils, as was the one analysed here (Cornforth, 2005; Rahardjo et al., 2007; Lu and Godt, 2008).

This event caused serious damage to different buildings. Regarding the damage caused, in the initial stretch of the slope (its head), a house was dragged and destroyed and another was seriously damaged. On the right bank of the mentioned house, another building was affected. In total, this event left a balance of two buildings destroyed and one seriously compromised. Although 15 people lived in these houses, there were no fatalities. About 20 houses were to be constructed at the head of the slope; fortunately, the event happened before this construction. Figure 7 shows an aerial picture from 2006 before the disaster as well as the affected area and landslide in 2010.

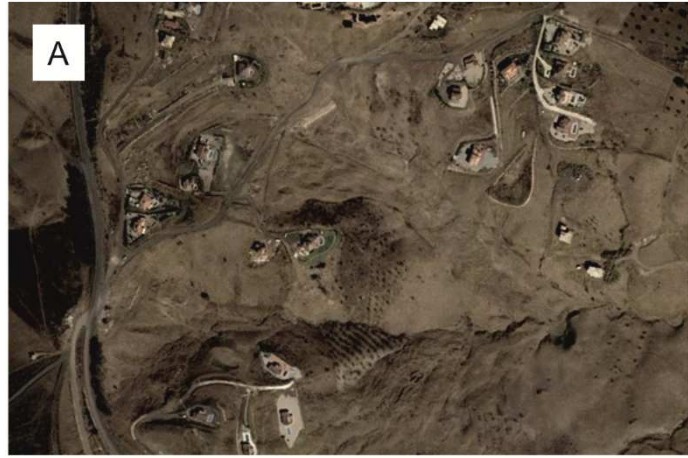

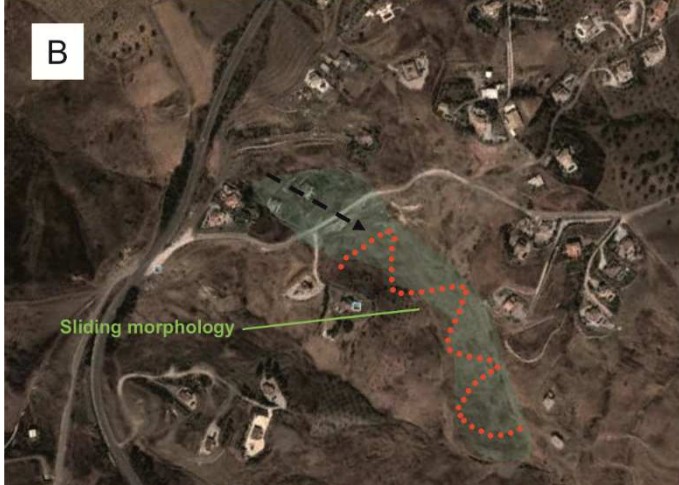

Gutiérrez-Martín, A.

***Figure 7.*** *A) An aerial photograph from before the event (2006). B) An aerial photograph taken after the*
*landslide (2010).*

**4.2 Event features and geometry**
For this example, we used data of IGN, the Spanish National Geographic Institute
(http://centrodedescargas.cnig.es/CentroDescargas), and downloaded bit map MTN25, that is
a 1:25000 topographic map in ETRS 89 coordinates and UTM projection. The downloaded map
is generated in a file by means of a geo-referenced digital rasterization (vector to raster
conversion). Specifically, we downloaded page number 1039, which is the one corresponding
to the landslide zone of the case study. Figure 8 shows the area of the case study.

From this map we obtained the topographic information to acquire all necessary profiles to
study the landslide. Moreover, as our algorithm is a 2D model, with this topographic map we
study the critical curve of the slip in the most unfavourable profile of the landslide (Figure 8).

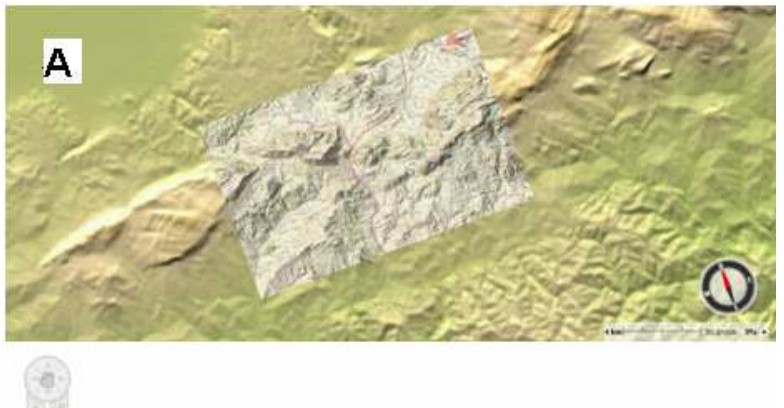

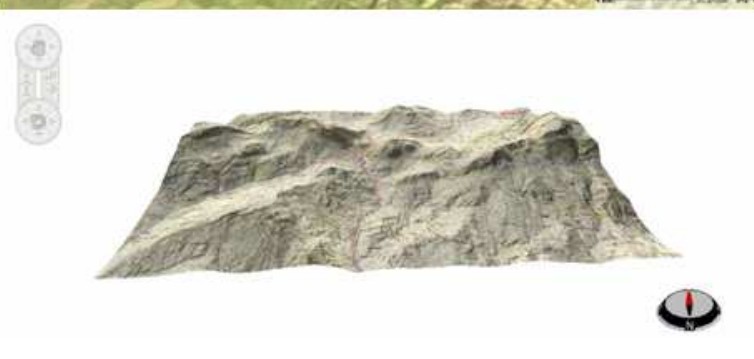

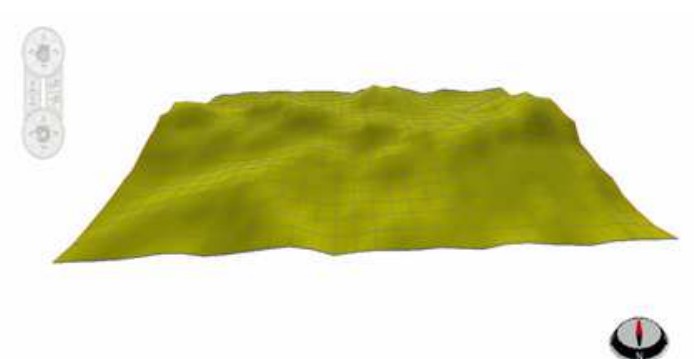


Gutiérrez-Martín, A.

**Figure 8**. *A) Topographic map in a GIS map; page number 1039 of the IGN (Spanish National Geographic*
*Institute).*

It is well known that mass movements, such as landslides, are highly complex morphodynamic
processes. We selected The Hundidero as our study area because it is prone to landslides. In
order to analyse this case study using our model, we first calculated the initial displaced
volume of the study area. According to the dimensions of the problem, the initial displaced
volume was calculated, equivalent to the volume of half an ellipsoid (Varnes, 1978; Beyer,
1987; Cruden and Varnes, 1996) that is Vol = 1/6 π (width x length x depth). In our particular
case, the width was equal to 70 m, the length equal to 235 m and the depth equal to 5 m,
making up a total volume of 4.364 m$^3$ (Figure 9). Taking an average of 33% elongation, as
proposed by Nicoletti and Sorriso-Valvo (1991) and Cruden and Varnes (1996), we determined
that the total material displaced in this landslide had an approximate volume of 5.804 m$^3$. In
this mass displacement, it is also necessary to consider material added by erosion and dragged
from the initial mass displaced. In Figure 7, the straight line indicates the first rotational
movement, and the zigzag line shows the planar drag and glide after the first rotational
movement. The green region is the total area displaced or affected by mass movement. After
the first circular movement, the mass moved rapidly, associated with a continuous rise in
incremental pore pressure and the rapid reduction of shear strength, without allowing
pressure dissipation.

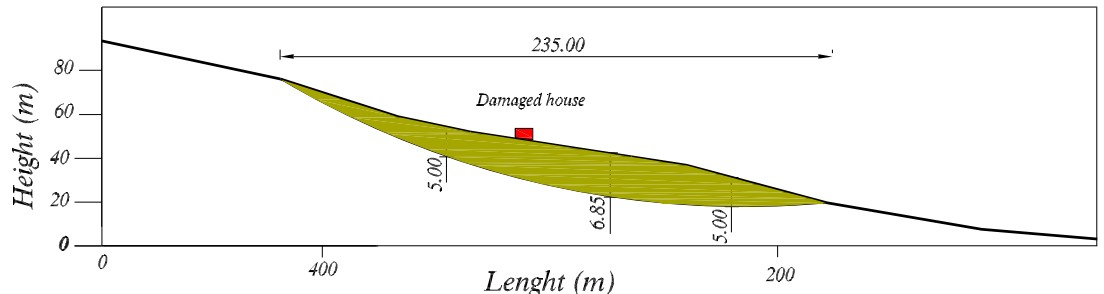


**Figure 9**. *Characterisation and longitudinal section of the rotational sliding (Geolen S.A., 2010). The*
*location of the dragged house is noted in red: Analysed by the TS model.*
The initial spit of land had an approximate size of 235 m in length by 70 m in width. Due to this
initial displacement, there was a drag and a huge posterior planar displacement of about
550 m length, affecting a zone with several parcels of land and buildings. These sizes were
confirmed using aerial photography and field data. The soil is basically composed of clays of
variable thicknesses, of fine grain, with fluvial sediments and silty clay. The authors obtained
this data by conducting a field survey, as well as through the laboratory tests carried out by the
laboratory Geolen S.A. (Geolen, 2010). From a geological and geotechnical point of view,
according to a survey of those present as the laboratory extracted the materials, different
lithological levels can be distinguished, as shown in Table 1.
**Table 1.** *Lithology of the area affected by the failure, according to the laboratory tests of*
*Geolen S.A. No groundwater level was detected.*

| Level/layers | Lithology | Depth (m) |
|---|---|---|

Gutiérrez-Martín, A.

| | | |
|---|---|---|
| **LEVEL 1** | Silty sand with natural schistose pebbles | 0.90 |
| | Silty clay with marl intercalations | 4.20 |
| **LEVEL 2** | Colmenar unit, upper oligocene–lower miocene | |
| | Sandy clay | 9.00 |
| **LEVEL 3** | Colmenar unit, upper oligocene–lower miocene | (*end of the probe*) |

The laboratory tests included a sieve analysis (following UNE 103 101) in three of the samples
extracted from the field, at depths of 1.80–2.00 m, of which 70.3% was composed of clay and
silt; according to this, the sample is classified as cohesive. The liquid limit and the plastic limit
were determined on two of the samples (following UNE 103 103 and UNE 103 104,
respectively), yielding liquid limit values of 57.5% and 64.2% and a plasticity index of 37%,
respectively. According to the lab results, the material can be classified as high plasticity
material with the potential of having a high water content. The landslide analyzed began in
February 2010, ending in March of that same year. However, based on the field inspection and
the analysis of the rainfall series in the La Viñuela region in 2010 (see Figure 10), it can be
inferred that the main causes of the event were:
• The poor geomechanical parameters of the material that formed the affected hillside,
and
• The hydrometeorological conditions in the days preceding and days after the event,
according to the histogram.

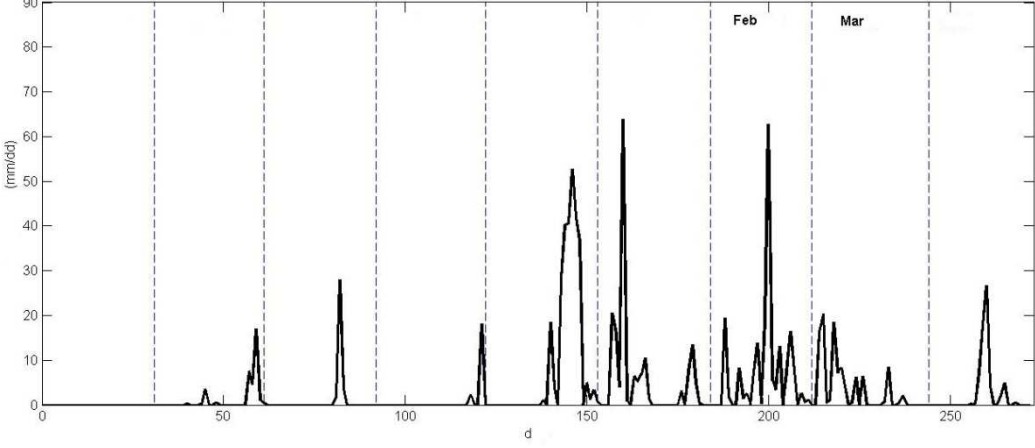


*Figure 10. Rainfall histogram at La Viñuela from August 2009 to April 2010. Rainfall data has been*
*provided by the Spanish Meteorological Agency (Station of Viñuela).*
Most of the landslides observed during these days occurred as a consequence of exceptionally
intense rainfalls. The precipitation data was provided by the meteorological station of La
Viñuela (Figure 10). It can be observed that large amounts of precipitation fell during the
months of December, January, February and March of 2010, with peaks of most 60 l/m2 in a
single day (January and February). In total, 890 l/m2 fell in the 2009-2010 hydro cycle, which
ended at the end of April 2010. This is a key point in slope stability to consider when dealing
with areas capable of having high infiltration rates.

Gutiérrez-Martín, A.

The rotational slide analysed had occurred between level 2 and level 3, when the water
content reached that depth, as confirmed by the infiltration calculations in the terrain (see
graphs in Figure 9, reaching depths of up to 5 m). Two direct shear tests (consolidated and
drained) were conducted in unaltered samples extracted from the boreholes at 3.00–3.60 m
and 4.00–4.60 deep. The cut-off values of the soil are specified in Table 2. Those values were
used in the developed software to obtain the safety coefficient and the theoretical failure
curve.
**Table 2.** *Summary chart of the characteristics of the soil analysed at the GEOLEN S.A. laboratory: $\phi$ the*
*angle of internal friction, c the cohesion, $\gamma_{Sat}$ the saturated specific gravity and $\gamma_a$ the apparent specific*
*gravity.*

| Soil parameter | Result | Units |
|---|---|---|
| $\phi$ | 17 | º |
| C | 0.27 | $N/mm^2$ |
| $\gamma_{Sat}$ | 2000 | $N/mm^3$ |
| $\gamma_a$ | 1650 | $N/mm^3$ |


The dynamic and continuous tests were carried out by the Geolen S.A. laboratory with an
automatic penetrometer ROLATEC ML-60 A type. The data obtained was transcribed by the
number of strokes to advance the 20 cms tip, which is called the "penetration number" ($N_{20}$).
This test is included in the ISO 22476-2:2005 standard as a dynamic probing super heavy, and
consists of penetrating the ground with a conical tip of standard dimensions. The depth of the
failed mass can be estimated, as well as the theoretical failure curve for an increase in the soil
consistency (see data in Table 3).
The change in the geomechanical response of the soil takes place at a depth of 4–5 m,
according to the results of $N_{20}$ and US (samples without changes) taken along the analysed
column. In this case, the sloped ground mass showed a characteristic striking relationship of a
displaced terrain (Gonzalez de Vallejo et al., 2002). This differs from the underlying or
unmoved terrain, which indicated a more consistent striking relationship that was taken within
the area of the landslide behind the house drawn in accordance with the analysis of the hits
$N_{20}$ from Table 3.
**Table 3.** *Summary chart of the soil analysed at the GEOLEN S.A. laboratory. Bold values show, according*
*to the data of the field penetrometers, the depth mobilized by the rotational sliding.*

| Depth (m) | Hits $N_{20}$ | Consistency | Admissible stress ($N/mm^2$) |
|---|---|---|---|
| **0.00 – 1.00** | **4** | **Soft** | **0.03** |
| **1.00 – 2.00** | **3** | **Soft** | **0.02** |
| **2.00 – 3.00** | **6** | **Slightly hard** | **0.04** |
| **3.00 – 4.00** | **7** | **Slightly hard** | **0.05** |
| **4.00 – 5.00** | **10** | **Slightly hard** | **0.07** |
| 5.00 – 6.00 | 19 | Moderately hard | 0.12 |
| 6.00 – 7.00 | 52 | Hard | 0.31 |
| 7.00 – 8.00 | 63 | Hard | 0.35 |
| 8.00 – 8.60 | 84 | Hard | 0.44 |


Gutiérrez-Martín, A.

**4.3 Input data**
To analyse the topography of the critical section, we obtained the DEM data from ArcGIS 10
software programme (Esri, 2010), with a scale of 1:1000, through Spanish National Geography
Institute (IGN) raster maps, with adequate accuracy. These data were interpolated to a 2 m
grid using a triangulated network interpolation methodology. Orthophotos proved very useful
to locate the landslide with accuracy and to validate the field survey. The model developed
here applies to failure in an initiation zone, in addition to predicting landslides, including those
induced by the infiltration of critical rains.

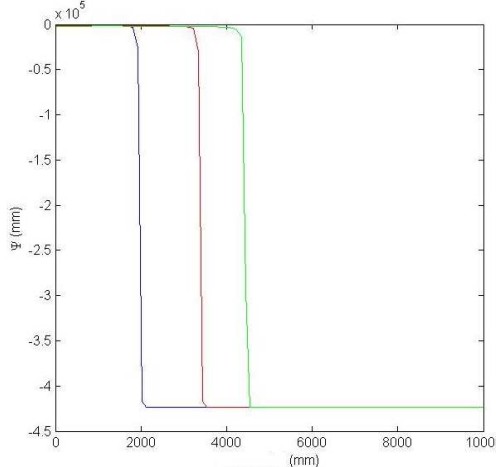 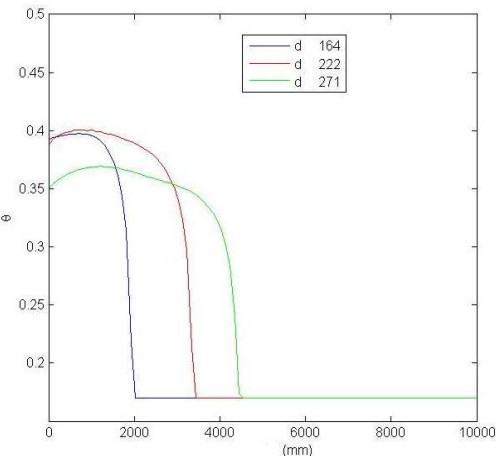


***Figure 11.*** *Left: hydraulic potential. Right: volumetric water content. Both have been plotted as a*
*function of the depth (mm) at different times (d).*
To complete the input data, we plotted the hydraulic potential and the volumetric water
content, as a function of depth in the ground for different time steps, using a previously
developed infiltration model, as shown in Figure 11 (Herrada et al., 2014). The figure shows
the evolution of how the wetting front advances can be observed. These reached almost 5 m
deep at the end of April 2010.

**4.2 Analytical results**
We applied the TS model using topographic data obtained from the ArcGIS 10 software
program. We did so to obtain the degree of stability of the sliding land based on the angle of
internal friction, the cohesion, the density and the angle of the slope we analyzed. Figure 9
shows the analytical results from the real slope, by studying and analyzing the most
unfavorable profile of the landslide studied. In addition we compared the results given by the
developed TS model and the results given by STB 2010 model, using free surfaces in both
cases. In our model the worst curve (shown in green) was calculated automatically from the
initial curve (show in blue), resulting in $F_S$ = 2.300, in the dry state (Figure 12).

Gutiérrez-Martín, A.

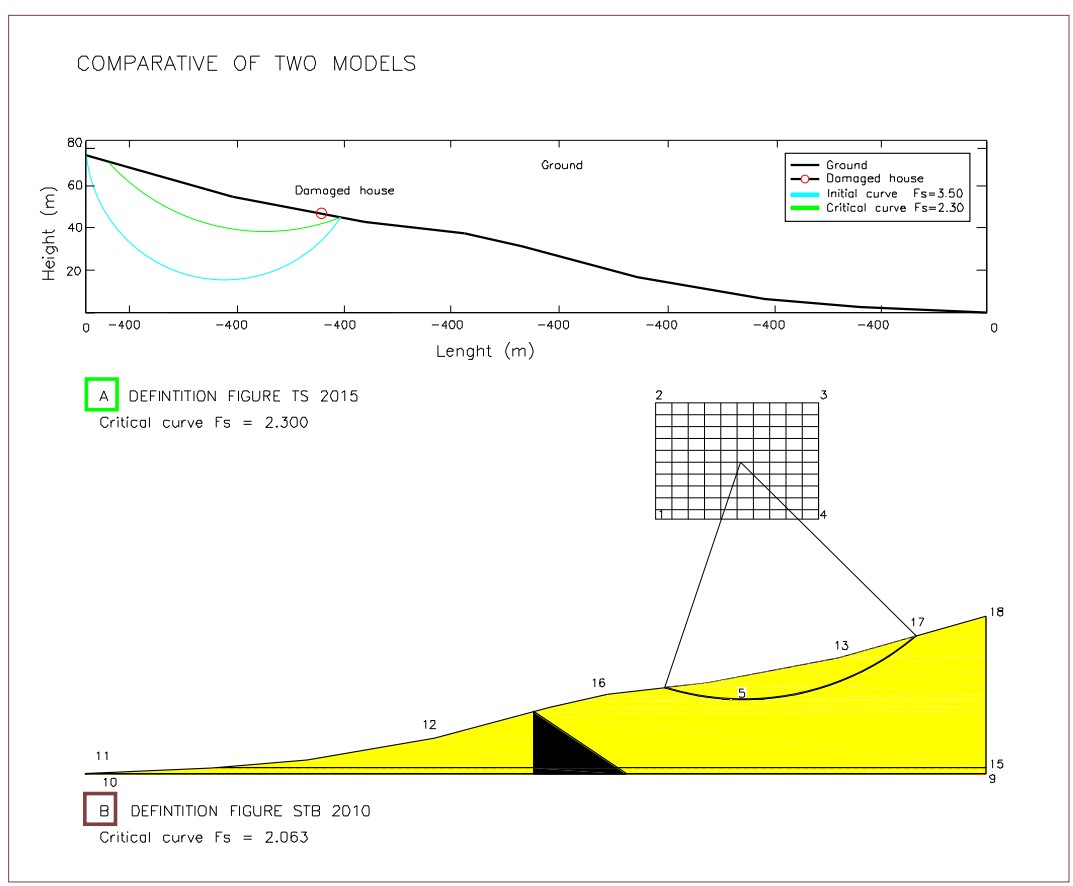


**Figure 12.** *Top: TS model with a critical failure of $F_S$ = 2.300. Bottom: results from the STB 2010 model with an Fs of 2.063.*

As can be noted, the failure curves are similar, and the safety coefficients $F_S$ only differ by 0.237. In both cases, the results indicated are conservative estimates, resulting in a stable slope that was not realistic, as was the case in La Viñuela. In order to get the most unfavourable curve, which would match the analysis of the actual event, the pore coefficient must be introduced. At the first runs of the model, the $r_u$ was equal to zero (dry soil – Figure 9), but if this value is changed to $r_u$ = 0.35, the results are quite different (Figure 13). The resulting failure was near the surface and the top cut with the slope found relatively near the houses. Taking into account the infiltration of rainwater, the slope analysed in the TS model showed a value of $F_S$ = 0.98, in other words, that it was unstable.

This calculation and the theoretical failure curve provided by our model was able to reproduce, in a realistic way, the landslide which occurred in La Viñuela. Our model found that the critical surface area that corresponded with the profile of the terrain was 12.927.45 m$^2$, which closely matches the real situation. In the STB 2010 programme, it was 7.825.35 m$^2$; therefore, our prediction was more accurate.

Gutiérrez-Martín, A.

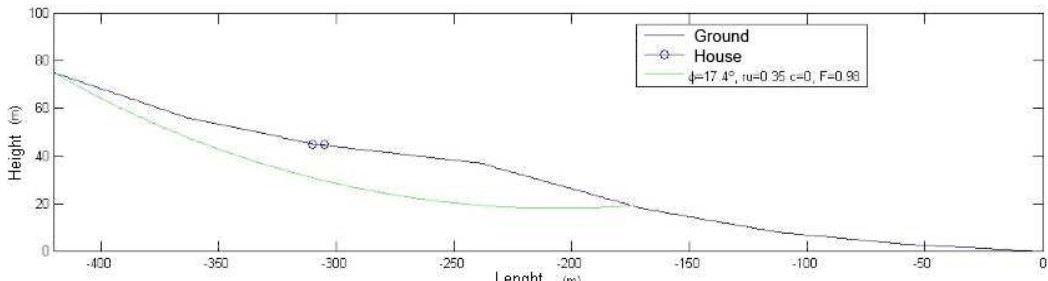

**Figure 13.** *A new calculation including the pore coefficient $r_u$ showing the worst curve in green. The circles show the houses dragged by the landslide.*

As mentioned, the STB 2010 model does not allow stability calculations to apply to rainfall infiltration on a hillside. Hence, it is not capable of predicting a hillside's instability in a critical rainfall scenario, which was critical in the slope analysed. The STB 2010 model found that the hillside studied had an Fs of $F_S$ = 2.063; that means it was a very stable slope. Consequently, our original algorithm TS model appears to be more efficient and accurate.

If we compare the results of the penetrometric tests (Table 3) and the laboratory tests (Geolen 2010) summarized in the actual critical surface in the most unfavourable profile of landslide (Figure 9), with those offered by our algorithm TS (Figure 13) to which we apply the infiltration factor $r_u$=0.35, (high interstitial pressure) we can check the similarity between the two critical surface of the landslide.

A value of $r_u$ = 0.35 has been introduced in the calculation and the code gave us a value of the slope safety factor of Fs = 0 .95 (unstable), when in the dry state the code calculated a safety factor of Fs = 2.300 (stable). The calculation of the safety factor in the STB 2010 program; that lacks the analysis of infiltration in the calculation, offered a result of Fs = 2.063 (stable).

Using the STB 2010 program, we would not have been able to previously detect the landslide of the case study of the paper, calculation that is not normally done in the stability calculations; with the calculation with our code we could have avoided the collapse of the building.

With these results, The Terrain Stability analysis performed using the developed model defines fairly well the slip-breaking curve that intuitively appears to be susceptible to failure, especially when heavy rains occur. As an example, the landslides which occurred in the La Viñuela area could only have been predicted if the infiltration had been taken into account. Even then, it could not have been done with other available software programmes, which were not able to consider it.

## 5.    Conclusion

The terrain stability (TS) analysis defines fairly well the critical surface to landslide in 2D of each profile of the analyzed slope and the safety slip factor (Fs). We developed this model due to the need for a useful tool to predict landslides, especially when heavy rains occur.

The TS model we developed uses the Spencer's method, which is more precise than the modified Bishop method, model used by other software such as the case of the STB 2010, so it

Gutiérrez-Martín, A.

differs in the results it provides for the $F_S$. It also takes into account the factor of water
infiltration due to critical rains, which other software programmes do not consider. A failure
surface can be determined by constraints using the MATLAB function fmincon. The data
needed to run the model include soil and climate properties that may vary in space and time.
The exit indices of the analysis ($F_S$) should be interpreted in terms of relative risk. The methods
implemented in the TS model are based on data structures, which are based on the data entry
of the elevation model (DEM), so we obtain a topographic map, a key element to obtain the
topographic profile to be studied with our algorithm.
In the case study analysed, the slope was initially stable and was so determined by the analysis
performed with the STB 2010 model. However, the slope became unstable due to the heavy
rains of that hydrological period, which called for the application of the pore pressure
coefficient $r_u$. For analysing cases of heavy rain, this model is a powerful tool for determining
slope stability. In addition, thanks to the great versatility of this model, it is applicable to any
analysis in other parts of the world, based on the methods of limit equilibrium (Spencer, 1967).
The TS model can also be used in combination with GIS software, SINMAP, TRIGRS model and
aerial photographic analysis, as well as mapping techniques or even as part of other models
like the coastal recession models (Castedo et al., 2012).

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

Gutiérrez-Martín, A.