# Peer review of "Development and validation of the Terrain Stability model for assessing landslide instability during heavy rain infiltration."

_Natural Hazards and Earth System Sciences, 2018_

## Referee Comment (RC1) · Anonymous Referee #1 · 28 Aug 2018

[referee-annotated manuscript omitted]

---

## Referee Comment (RC2) · Anonymous Referee #2 · 4 Sep 2018

The manuscript describes a numerical approach to slope stability, and the corresponding original software. The model is two-dimensional, and its applicability is limited to a single slope; advantages are the software being freely available and inclusion of wet soil conditions, apparently missing in existing commercial software.

I believe that the manuscript suffer from several limitations, and in my opinion is not suitable for pubblication in NHESS. I will try and motivate my opinion in three different sections, as requested by NHESS reviewing guidelines.

______________________ General comments, main issues:

I believe that the material in the manuscript is organized in a rather confusing way, and

that key sections of the text do not contain the information they are supposed to.

The Title suggests that the paper deals with landslide "risk", while it describes a numerical model for slope stability assessment. The generally accepted definition of "risk" associated with a natural hazard is the product, or the combination, of the likelihood of an event of the given kind ("hazard") and "exposure", or "vulnerability", of human life and infrastrucuture to that kind of hazard. Morevoer, the generally accepted definition of "hazard" is, in turn, the product of spatial probability, temporal probability and magnitude of an event of the given type to occur. The model described in the manuscirpt deals with spatial and magnitude assessent of landslides; it is not clear to me whether a temporal component is included. Surely we cannot speak about "probability" here, because the model obtains a factor of safety, which is clearly NOT a probability. In order to obtain a probabilistic interpretation of the factor of safety, one needs to perform additional, non trivial steps. See, for example:

- S Raia, M Alvioli, M Rossi, RL Baum, JW Godt, F Guzzetti (2014). Improving predictive power of physically based rainfall-induced shallow landslide models: a probabilistic approach. Geosci. Model Dev. 7 (2), 495-514. https://doi.org/10.5194/gmd-7-495-2014

- S Zhang, L Zhao, R Delgado-Tellez, H Bao (2018). A physics-based probabilistic forecasting model for rainfall-induced shallow landslides at regional scale. Nat. Hazards Earth Syst. Sci., 18, 969–982. https://doi.org/10.5194/nhess-18-969-2018

- E Canli, M Mergili, B Thiebes, T Glade (2018). Probabilistic landslide ensemble prediction systems: lessons to be learned from hydrology. Nat. Hazards Earth Syst. Sci., 18, 2183–2202. https://doi.org/10.5194/nhess-18-2183-2018

Moreover, it is not true that the model itself includes an assessment of vulnerability, which must be taken into account separately and, most importantly, with additional (and often difficult to obtain) data. The Title also mention validation of the model, which was actually performed in a rather qualitative way. It also mentions the expression

"during heavy rain infiltration", which is not actually substantiated in the manuscript since, again, no explicit time dependence is implemented as the word "during" would suggest, and no actual "infiltration" is considered, but only its effective result - namely, an effective value for pore pressure calculated at an arbitrary depth under the soil surface. At least, this is what I can understand after reading the whole manuscript. I will give more details below.

The Abstract contains unnecessary information (the first two, long sentences), a few inaccuracies (see below) and, most importantly, fails to properly and succintly introduce the methods, results and conclusions obtained in the manuscript. A GIS support is mentioned, while the whole code is implemented in Matlab.

From the Introduction, we understand that the scope of the proposed model is slope stability from the engineering point of view, which is a perfectly legitimate field for a NHESS pubblication. Nevertheless, given the range of expertise that the Journal is devoted to, I believe that the topic should be put in a broader perspective. The Authors made explicit reference to a number of commercial software programs that are supposed to have the same applicability domain. I believe that the existence of other, well-known models for slope stability assessment with a broader applicability domain should be acknowledged, and the relationship between these models and engineering of individual slopes should be elucidated. The following list of a few such models is certainly not exhaustive but it is a starting point:

TRIGRS:

- RL Baum, WZ Savage, JW Godt (2008). TRIGRS a Fortran program for transient rainfall infiltration and grid-based regional slope-stability analysis. US geological survey open-file report 424, 38 https://pubs.usgs.gov/of/2008/1159/

- M Alvioli, RL Baum (2016). Parallelization of the TRIGRS model for rainfall-induced landslides using the message passing interface. Environmental Modelling & Software 81, 122-135 http://dx.doi.org/10.1016/j.envsoft.2016.04.002

[Figure]

SINMAP:

- RT Pack, DG Tarboton, CN Goodwin, (2001). Assessing Terrain Stability in a GIS using SINMAP. In: 15th annual GIS conference, GIS 2001, Vancouver, British Columbia, February 19-22. (and references therein; SINMAP is actually referred to at the very end of the paper, without any description or reference)

SHALSTAB/r.shalstab: -http://calm.geo.berkeley.edu/geomorph/shalstab/index.htm

-https://grass.osgeo.org/grass74/manuals/addons/r.shalstab.html

GEOtop/GEOtop-FS:

- R Rigon, G Bertoldi, TM Over (2006). GEOtop: A distributed hydrological model with coupled water and energy budgets. Journal of Hydrometeorology 7 (3), 371-388 https://doi.org/10.1175/JHM497.1

- S Simoni, F Zanotti, G Bertoldi, R Rigon (2008). Modelling the probability of occurrence of shallow landslides and channelized debris flows using GEOtop-FS. Hydrol Processes;22(4):532-545. https://doi.org/10.1016/j.proeps.2014.06.006

r.slope.stability:

- M Mergili, I Marchesini, M Rossi, F Guzzetti, W Fellin, (2014). Spatially distributed three-dimensional slope stability modelling in a raster GIS. Geomorphology 206: 178-195. http://doi.org/10.1016/j.geomorph.2013.10.008

Moreover, in the Introduction, the Authors state that stability models are limited to 2D approaches, while a few examples exist of 3D models. For example, among others:

- M Mergili, I Marchesini, M Alvioli, M Metz, B Schneider-Muntau, M Rossi, F Guzzetti (2014). A strategy for GIS-based 3-D slope stability modelling over large areas. Geoscientific Model Development 7 (6), 2969-2982 http://doi.org/10.5194/gmd-7-2969-2014

- ME Reid, SB Christian, DL Brien, ST Henderson (2015). Scoops3D—software to analyze three-dimensional slope stability throughout a digital landscape (Version 1.0). Virginia: U.S. Geological Survey

- TV Tran, M Alvioli, G Lee, HU An (2018). Three-dimensional, time-dependent modeling of rainfall-induced landslides over a digital landscape: a case study. Landslides, 1-14 http://doi.org/10.1007/s10346-017-0931-7

About the resources required to apply stability models on large areas, these papers describe methods, including parallel computing, to cope with such issue:

- M Mergili, I Marchesini, M Alvioli, M Metz, B Schneider-Muntau, M Rossi, F Guzzetti (2014). A strategy for GIS-based 3-D slope stability modelling over large areas. Geoscientific Model Development 7 (6), 2969-2982 http://doi.org/10.5194/gmd-7-2969-2014

- M Alvioli, RL Baum (2016). Parallelization of the TRIGRS model for rainfall-induced landslides using the message passing interface. Environmental Modelling & Software 81, 122-135 http://dx.doi.org/10.1016/j.envsoft.2016.04.002

Sections 2 and 3, devoted to a description of the methodology implemented in the model, are confusing, and I cannot understand what are the assumptions and the relevant details of the method implemented in the software, and whether it is a novel enough approach. I will give more details later on.

Section 4.3 is devoted to the description of the results obtained using the proposed model. This section is very confusing, again. I believe that the comparison of the results of the proposed model with the another model, and with a real landslide scenario, are presented in an unsatisfactory way, since they are qualitative almost everywhere and it is difficult to understand what the quantitative comparisons refer to. Moreover, there is a large fraction of text which does not pertain to results but to the methodology itself.

Eventually, in Section 5, devoted to describe conclusions of the manuscript, again I do not find enough evidence of actual conclusions drawn from the results. In addition to repeating already mentioned concepts in a, in my opinion, misleading way (i.e., use of "prediction", of "time", etc.), there are a couple of expressions which, I believe, are not allowed in assessing the conclusions in a scientific paper. First, the Authors state that the proposed model "defines fairly well areas that intuitively appear to be susceptible to landslides and defined rigorously the failure curve". In this sentence, "fairly well" and "intuitively" are not good enough to assess the predicting performance of a quantitative model. Moreover, the "rigorous" definition of slip surfaces does not appear to be substantiated by the presented results, as I will explain at lenght in the following. Then, the expression "this model is probably the most powerful tool for determining slope stability", is again not substantiated by the presented results. Eventually, a reference to the SINMAP model comes out of the blue, in the second-last line, which is unjustified.

_____________________ Other specific comments:

In the Abstract, in addition to unrelevant information in the first two sentences already mentioned above, I believe that a few other ambiguities exist. It is stated that "Climate is one of the main factors [affecting slope stability, Ed.], especially when large amounts of rainwater are absorbed in short periods of time". The paper does not discuss climate effects on landslides, or correlations between the different factors determining the climate of a given region and landslides. Thus, this should not appear in the Abstract, which must contain a short description of the specific topic discussed in the paper; maybe in the introduction, if a sufficiently clear link is made with the topic of the paper. A quantitative relationship with climate (actually, climate change), rainfall events and slope stability including actual time-dependent account for rainfall infiltration can be found in, e.g.:

- S Gariano, F Guzzetti (2016). Landslides in a changing climate. Earth-Science Reviews, 162, 227-252. https://doi.org/10.1016/j.earscirev.2016.08.011

- M Alvioli, M Melillo, F Guzzetti, M Rossi, E Palazzi, J von Hardenberg, MT Brunetti, S Peruccacci, (2018). Implications of climate change on landslide hazard in central Italy Science of the Total Environment 630, 1528-1543. https://doi.org/10.1016/j.scitotenv.2018.02.315

The Authors claim that the model is "supported by a GIS", which is not true. The only step in which a GIS can (can) be used is when they mention that the terrain profile was obtained fom a DEM. The model is coded in Matlab, and not within a GIS. The statement "the model is especially useful for predicting .. scanarios of heavy unpredictable rainfall" is very bold. How can accout for rainfall into a mathematical model, if the rainfall is unknown?!?

In the Introduction, in addition to what I have written above. At lines 35-36, the Authors refer to a 2002 paper commenting "nowadays"; I believe that more recent papers exist, other that a 16-years-old one. FOS is used but not defined. Moreover, throughout the Manuscript, the Authors refer to factor of safety using both "FOS" and "Fs", apparently for no good reason.

Section 2, where details of the models are described, is rather confusing to me. After the relevant equations are introduced (missing the definition of a few quantities here and there), it is stated that these coupled equations must be solved to obtain the factor of safety Fs and the angle theta. Then, theta is assumed as constant, for no apparent reason, other than the sentence that "it provides optimal results", with no further explanation or justification. The whole meaning of theta should be explained in a better way, in my opinion. Then, the role of pore pressure is introduced. I believe it is imposisble to understand why they Authors refer to a "pore pressure distribution" and then use a single value (effective coefficient?) for it, or if they actually use a distribution. Moreover, it is impossible to understand if a time dependence - according to actual rainfall infiltration as a function of, well, rainfall intensity and varying water content in the soil, was taken into account or not.

Line 120, "precise" should be "accurate"

Lines 125-126, "lower than 1, or stable if it is higher than 1" miss the (mathematical) possibility that Fs=1. Moreover, the statement that Fs "tend to be higher than one" deserves an explanation in addition to Burbano et al. (2009), since it represents the whole point of the paper.

Lines 133-136: I do not understand the seentence from "However" to "exhogenous factors of the slope". Moreover, as for the sentence "Fs>=1.3 can be considered stable by most standars", please se my detailed discussion below.

Lines 141-142: whay does the initial curve depend on the "data introduced"? Does the user specify the whole curve, or what? What is the dependence of the results upon such an initial, arbitrary choice?

Section 3 tries to describe in somewhat more detail the operation of the software, but this is also done in a confusng way, in my opinion. First, I find confusing to name a terrain stabilitt model as "terrain stability", but this might be my personal taste.

Figure 2 shows a terrain profile, obtained from a DEM, and it is mentioned that the profile was splitted into 500 slices. How does the profile emerge from the DEM? Which DEM, with what resolution? Also, why 500 slices? Is this the number of DEM cells along the profile, or is it less, or more? If it is less, why is it so? If it is more, how do we interpolate the DEM and why? Is the use of a circular shape for the slip surface a limitation, which I believe it is, given that other engineering-like models use "trial" (and not "first", since there is no hierachy in the different "trial" surfaces) surfaces of any shape? For example, in the SSAP model, which has apparently an applicability domain similar to the model presented in the manuscript and it is free as well, slip esurface can be of any shape, to my knowledge; I might be mistaking: https://www.ssap.eu.

Fonts are way too small in any figure in the Manuscript.

The introduction of wet conditions seems to be performed as a separate step, is it true?

This is relevant, and really hard to understand. Why is the modification only considered "at the basis" of the terrain slice? This would account for a modified shear stress at the bottom of the slice, but where is the contribution of water weigth along the whole slice? Is this approach rigorous, or is it an approximation? The curves in Fig. 3 are described to be different because of the "different data introduced": what kind of data did the Authors change? Introducing pore pressure effect is not "different data", it is a different physical mechanism, thus a different model model. The statement "after the outcome here, it can be stated that the rainwater infiltration factor is necessary to predict instabilities of the slope" contains two logical mistakes, in my opinion. Firstly, for a "prediction" to be performed, one needs to have spatial and time dependence, or at least specifiy what is it that one is trying to predict, which is not the case here. Secondly, to establish that infiltration is a "necessary" factor, it is not enough to show that results with and without inclusion of the pore pressure correction are different: one must show that the case with inclusion is closer to reality than the other case!

Lines 247-249: I do not understand the sentence "if this infiltration factor is small enough, taking into account the safety coefficients, the design may still be adequate, but there was a lack of critical information for calculating this parameter" is not only difficult to understand, it also poses severe doubts on how is it possible to develop/test a model in which rainfall infiltration is supposed to be one of the key ingredients, and then the test case is taken as one in which not enough data exist to apply the model itself??

Line 300: "dimensions" shold be "sizes", or something of the like.

Information in lines 320 to 325 seems to be trivial enough not to be highlighted with a bulleted list. Moreover, the statement ".. after the event, accodring to the histogram" is rather misterious, since I can't find any event in Fig. 7.

In Section 4.2, Figure 7: when did the landslide considered in the paper occur, in the timeline? This is relevant information, is it not? The Authors refer to "level 2 and level

3": what are the levels the Authors refer to? They also refer to "infiltration calcula-tions", when and how did they perform the mentioned calculations? This is probably described in Section 4.3, but this comes out of the blue and I do not understand how the calculations were done, and why they were not described in the methodological Section, instead of the "input data" Section.

Section 4.3 is devoted to describe the results obtained using the proposed model with a real landslide scenario. This section is very confusing, again. First of all, the Authors compare theirs results with the results obtained from a different model/program; so far, so good - even if this should have been mentioned briefly in the Introduction and/or methology sections, since it is part of the reseach method. Then the refer to "previous calculations", about which the reader is not aware, and they discuss curves that are non existing in Figure 9 (yellow and red curves?). They pretend that the "curves are similar", without any attempt to quantify the extent to which they are similar. Of course they are similar indeed, since all of them are circles arcs, but that does not seem to me to be enough, as a comparison. The same goes for the comparison with the real landlside failure curve, which I do not understand if it was actually measured or not, or if it is measurable at all. Then, the Authors refer to measures in square meters of the "surface area that corresponded with the profile", which I do not understand. What does "correspond" mean? The software is supposed to provide a two-dimensional failure curve on a vertical plane, there is no corresponding surface area. Or, at least, I don't see what it is, particularly I do not see what is the "real situation" the Authors refer to.

In the same Section, the Authors refer to a "very stable" slope as one with an Fs much larger than unity. I believe this is a conceptual mistake. A model in which slope stability is assessed with an Fs defined as the ratio of destabilizing forces to stabilizing ones, there is no such thing as "more stable". A slope, or a DEM cell, or a slice, is unstable if Fs<1, and stable otherwise. Different degrees of stability are not defined in the model, since no attempt whatsoever exist (in this and in similar models) to quantify

the sensitivity of Fs results to the large number of parameters and assumtpions utilized to obtain the result, nor to give a measure of the uncertainty. We do not know if an enormous rainfall would change Fs by a tiny bit or by a large amount, nor what is going to happen if an earthquake comes about. In other words, values of Fs different from the exact value obtained from the calculations do not have different degrees of probability, thus different degrees of stability are undefined within such a model. At least, if no further analysis is performed. Lastly, in the same Section, six points are listed, which contains methodological remarks and no results, As such, these do not belong in this Section, but to a previous one.

_____________________ Technical comments:

English seems fairly good, but I am no native English speaker.

---

## Referee Comment (RC3) · Anonymous Referee #3 · 30 Nov 2018

The aim of this work/manuscript is the development of a software for single slope stability. A case study and a comparison with another software is presented. In my opinion, the main originality of the paper is represented by the inclusion in the software of the infiltration effects, according to the lacking in other software slope stability based, but the used theory (Spencer's method) and the way to calculate the interstitial pressure on the slice base is well known. Therefore, the proposed model is not innovative and the authors should give more emphasis to the originality of the developed software, clarifying the advantages also in term of time simulation. I suggest the authors to include a block diagram of the software in order to explain better their algorithm from the user definitions to outputs/results. Moreover, a sensitivity analysis of the parameters is missing: I suggest for example to add some plot, e.g., the safety factor varying the interstitial pressure coefficient $r_u$, the center of failure curve, the number of slices, the density of soil, etc. To me, in conclusion, the paper needs some improvements and major revisions should be required.

**Specific comments:**

The **section 1 (Introduction)** must be expanded citing other works that develop/use stability model, e.g.:

Anderson MG, Howes S (1985). Development and application of a combined soil water-slope stability model. Q. J. Eng. Geol. London, 18: 225-236

Crosta GB, Frattini P (2003). Distributed modelling of shallow landslides triggered by intense rainfall. Natural Hazards and Earth System Sciences (2003) 3: 81–93

Iverson RM (2000). Landslide triggering by rain infiltration. Water Resources Research 36(7): 1897-1910

Lu N, Godt J (2008). Infinite slope stability under steady unsaturated seepage conditions. Water Resources Research, Vol. 44, W11404, doi:10.1029/2008WR006976

Rossi G, Catani F, Leoni L, Segoni S, Tofani V (2013) HIRESSS: a physically based slope stability simulator for HPC applications. Nat Hazards Earth Syst Sci 13(1):151–66

Wu W, Sidle RC (1995). A Distributed Slope Stability Model for Steep Forested Basins. Water Resour. Res., 31(8), 2097–2110, doi:10.1029/95WR01136

Then I would mention empirical model that relies on rainfall thresholds, for example:

Aleotti P (2004). A warning system for rainfall-induced shallow failures. Eng Geol 73:247–265

Caine N (1980). The rainfall intensity duration control of shallow landslides and debris flows. Geografiska Annaler. 62A, 1-2: 23-27

Govi M, Mortara G, Sorzana PF (1985). Eventi idrologici e frane. Geol. Appl. e Idrogeol. XX, 2: 359-375

Guzzetti F, Peruccacci S, Rossi M, Stark CP (2007). The rainfall intensity-duration control of shallow landslides and debris flows: an update. Landslides, Vol. 5:3-17, doi: 10.1007/s10346-007-0112-1

Martelloni G, Segoni S, Fanti R, Catani F (2011). Rainfall thresholds for the forecasting of landslide occurrence at regional scale. Landslides DOI: 10.1007/s10346-011-0308-2

Wilson RC, Jayko AS (1997). Preliminary maps showing rainfall thresholds for debris-flow activity, San Francisco Bay Region, California. US Geological Survey Open-File Report 97-745 F

Moreover should be cited papers in which other approaches/theoretical studies for landslide prediction are used (for triggering and/or propagation), e.g.:

Crosta GB, Imposimato S, Roddeman DG (2003) Numerical modelling of large landslides stability and runout. Nat Hazards Earth Syst Sci 3(6):523–538

D'Ambrosio D, Di Gregorio S, Iovine G (2003) Simulating debris flows through a hexagonal cellular automata model: SCIDDICA S3-hex. Nat Hazards Earth Syst Sci 3(6):545–559

Iovine G, Mangraviti P (2009) The CA-model FLOW-S* for flow-type landslides: an introductory account.Proceedings of the 18th World IMACS congress and MODSIM09 international congress on modelling and simulation, 13–17 July 2009, pp 2679–2685

Martelloni G, Bagnoli F (2014) Infiltration effects on a two-dimensional molecular dynamics model of landslides. Nat Hazards, 73(1):37–62

Martelloni G, Bagnoli F, Guarino A (2017) A 3D model for rain-induced landslides based on molecular dynamics with fractal and fractional water diffusion. Commun Nonlinear Sci Numer Simulat, 50:311–329

Patra AK, Bauer AC, Nichita CC, Pitman EB, Sheridan MF, Bursik M, Rupp B, Webber A, Stinton A, Namikawa L, Renschler C (2005) Parallel adaptive numerical simulation of dry avalanches over natural terrain. J Volcanol Geotherm Res 139:1–21

The **section 2 (Terrain Stability model development)** needs some corrections:

1) The meaning of some parameters is missing in the text, e.g., in the equation 3 $R$ is the radius of the curvature and $\alpha$ is the angle of the slope referred to each slice (I suppose); in the equation 7 $\gamma$ is the density of soil and $h$ is the mean height of slice (if the height is not constant). Please check these!

2) In my opinion is not clear how the pore pressure is calculated by means of equation 7., i.e., how is the interstitial pressure coefficient $r_u$ calculated (according to heavy rainfall event)? Then, how does the equation 8 (Mohr-Coulomb law), for the calculus of $u$, come into play? In the article of Spencer (Spencer, 1967), assuming a homogeneous pore-pressure distribution as proposed by Bishop and Morgenstern (1960), the mean pore-pressure on the base of the slice can be written just like the equation 7 that is used for the calculation of the safety factor (substituting expression of $u$ in equation 5). Please clarify the need of equation 8!

The **section 3 (Terrain Stability (TS) model behaviour tests)**, in my opinion, should be renamed **Terrain Stability (TS) algorithm and tests** adding these points:

1) I suggest to include a block diagram of the software in order to explain in detail your algorithm from the user definitions to outputs/results.

2) As sensitivity analysis of the parameters is missing, I suggest for example to add some plot, e.g., the safety factor varying the interstitial pressure coefficient $r_u$, the center of failure curve, the number of slices, the density of soil, etc.

**Line 206**: It is not "centre", but center. Please, check the paper if other typos are present!

Concerning the **section 4:**

**Line 415:** I would not say "our innovative TS model", but "our original algorithm".

**Lines 421-444**: I would add this part in the section 3 where is requested the explanation of the algorithm (software).

---

## Author Comment (AC1) · 10 Jan 2019

We maintain that the manuscript is well within the aims of NHESS. We provide in the attached supplement, the constestations and contributions to the questions raised by you.

Thanks for your comments

regard greeting

GENERAL COMMENTS

(1) This paper deals with the ability to predict a landslide failure curve and the slope

factor of safety with a terrain stability (TS) analysis. Overall it is written in a good English, but I believe it is not as innovative as claimed for considering rainfall infiltration in the calculation of the factor of safety. I suggest focus on the ability to well-predict the landslide failure curve and surface area.

(2) We appreciate your comments on our model and specifically the ability to predict well the slip failure curve and the area of the breakage surface. We continue to believe in the originality of our model, incorporating the rainfall infiltration in the calculation of the safety factor with a terrain stability, by means of the infiltration factor of the Spencer method and its treatment in the calculation model implemented in Matlab. The TS method is a simple, but versatile computational procedure that is suitable for a normal computer

In addition, we also re-focus the paper on the ability to well-predict the landslide failure curve and surface area. In the intro a new text has been added:

(3) "The new developed software is fast and accurate in resolution of landslide failure curve and surface area, including the infiltration effects."

(1) The literature of relevance has not been adequately cited. I recommend reviewing more methods for slope stability analysis.

(2) We have incorporated the references in the introduction section and in the specific comments, which we believe improve the application literature. In the revised final version of the paper, we include a brief description of the state-of-art in order to clarify the improvements of our work.

(1) Chapter 2 is very confusing and the paragraphs are disjointed. As is, it is not easily readable. In chapter 3, the test should be described more accurately and the center of the failure curves should be shown in, at least, the first figure (Figure 2).

(2) Chapter 2 has been improved by incorporating in specific comments clarifications and syntax improvements, the center has been incorporated in Figure 2 (a new figure

2 has been done).

(1) In chapter 4 there is the need to mention the date of the slide.

(2) The date of the landslide has been introduced, which we had not included in the text, but we had marked it in figure n. °7, in the histogram. See specific comments.

(1) Both chapter 3 and 4 miss the description of the calculation of the pore pressure parameter. Move lines 421-444 to chapter 2 because they describe the characteristics of the TS model compared to the STB 2010.

(2) In the specific comments section show the description of the calculation of the pore pressure parameter. We have clarified this question.

Following your suggestions, we have moved lines 421-444 to chapter 2 because they describe the characteristics of the TS model compared to the 2010 STB.

(1) Please, check references consistency with the journal guidelines. For other comments and technical corrections see the attached file.

(2) We have reviewed and verified the references of coherence with the guidelines of the journal.

Please also note the supplement to this comment:
https://www.nat-hazards-earth-syst-sci-discuss.net/nhess-2018-192/nhess-2018-192-AC1-supplement.pdf
* * *
[Figure]

**Supplement:**

Dear Authors,

You - as the contact author - are requested to individually respond to all referee comments (RCs) by posting final author comments on behalf of all co-authors no later than 13 Jan 2019 (final response phase) at: https://editor.copernicus.org/nhess-2018-192/final-response.

**Comments by Anonymous Referee #1 (nhess-2018-192-RC1)**
[Answers in blue]

**GENERAL COMMENTS**

**(1)    This paper deals with the ability to predict a landslide failure curve and the slope factor of safety with a terrain stability (TS) analysis**. Overall it is written in a good English, but I believe it is not as innovative as claimed for considering rainfall infiltration in the calculation of the factor of safety. **I suggest focus on the ability to well-predict the landslide failure curve and surface area.**

(2) We appreciate your comments on our model and specifically the ability to predict well the slip failure curve and the area of the breakage surface. We continue to believe in the originality of our model, incorporating the rainfall infiltration in the calculation of the safety factor with a terrain stability, by means of the infiltration factor of the Spencer method and its treatment in the calculation model implemented in Matlab. The TS method is a simple, but versatile computational procedure that is suitable for a normal computer

In addition, we also re-focus the paper on the ability to well-predict the landslide failure curve and surface area. In the intro a new text has been added:

(3) "*The new developed software is fast and accurate in resolution of landslide failure curve and surface area, including the infiltration effects.*"

(1)    The literature of relevance has not been adequately cited. I recommend reviewing more methods for slope stability analysis.

(2) We have incorporated the references in the introduction section and in the specific comments, which we believe improve the application literature. In the revised final version of the paper, we include a brief description of the state-of-art in order to clarify the improvements of our work.

(1)    Chapter 2 is very confusing and the paragraphs are disjointed. As is, it is not easily readable. In chapter 3, the test should be described more accurately and the center of the failure curves should be shown in, at least, the first figure (Figure 2).

(2) Chapter 2 has been improved by incorporating in specific comments clarifications and syntax improvements, the center has been incorporated in Figure 2 (a new figure 2 has been done).

(1)    In chapter 4 there is the need to mention the date of the slide.

(2) The date of the landslide has been introduced, which we had not included in the text, but we had marked it in figure n. °7, in the histogram. See specific comments.

(1)    Both chapter 3 and 4 miss the description of the calculation of the pore pressure parameter. Move lines 421-444 to chapter 2 because they describe the characteristics of the TS model compared to the STB 2010.

(2) In the specific comments section show the description of the calculation of the pore pressure parameter. We have clarified this question.

Following your suggestions, we have moved lines 421-444 to chapter 2 because they describe the characteristics of the TS model compared to the 2010 STB.

(1) Please, check references consistency with the journal guidelines. For other comments and technical corrections see the attached file.

(2) We have reviewed and verified the references of coherence with the guidelines of the journal.

**SPECIFIC COMMENTS**

**SECTION 0: ABSTRACT**

(1) These sentences seem unjointed. Please, try to make them more fluent for reading.

(2) The comment seems right to us and we introduce the change. In the original document line 23-26.

(3) *"This model is especially useful for predicting the risk of landslides in scenarios of heavy unpredictable rainfall. We have called it (TS) Terrain Stability and programmed in MATLAB, which it allows us a simulation of the slope stability in a 2D spatial distribution. As originality in our algorithm a hydrological assumption has been incorporated in steady-state."*

(1) Be more specific: critical rainfall, critical point, critical surface..., I would suggest a keyword that refers to the model, such as: numerical model, 2D model, limit equilibrium model,..

(2) The comment seems right to us and we introduce the change.

(3) *"Keywords: Landslides, critical rainfall, limit equilibrium model, 2D model, critical surface."*

**SECTION 1: INTRODUCTION**

**(1)** I believe you should start the introduction paragraph with a general description of the landslide hazard, as you did in the abstract.

After that, you may depict what is a stability analysis and its evolution over time.

(2) The comment seems right to us and we introduce the change.

(3) *"Landslides, one of the natural disasters, have resulted into significant injury and loss to the human life and damaged property and infrastructure*

*throughout the world (Crozier and Glade, 2005; Dai et al., 2002; Parise and Jibson, 2000; Varnes, 1996).*

*Normally, heavy rainfall, high relative relief and complex fragile geology with increased manmade activities, have resulted in increased landslide (Gutiérrez-Martin, 2015). It is essential to identify, evaluate and delineate landslide hazard prone areas for proper strategic planning and mitigation (Bisson et al., 2014). Therefore, to delineate landslide susceptible slopes over large areas, landslide hazard zonation (LHZ) techniques can be employed (Fall et al., 2006; Casagli et al., 2004; Guzzetti et al., 1999; Anbalagan, 1992).*

*Landslides are resulted because of intrinsic and external triggering factors. The intrinsic factors are mainly; geological factors, geometry of the slope (Wang and Niu, 2009; Ayalew et al., 2004; Anbalagan, 1992; Hoek and Bray, 1981).*

*The external factors which generally trigger landslides are rainfall (Dai and Lee, 2001; Collison et al., 2000; Anderson, 1985). Several LHZ techniques have been developed over the past and these can be broadly classified into three categories; expert evaluation, statistical methods and deterministic approaches (Canili et al., 2018; Zhang et al.; 2018; Lari et al., 2016; Raia et al., 2014; Rossi et al., 2013; Lu and Godt, 2008; Fall et al., 2006; Casagli et al., 2004; Crosta and Frattini, 2003; Inverson, 2000; Guzzetti et al., 1999; Leroi, 1997; Wu and Sidle, 1995). Within these models, we want to highlight the empirical models that are based on rainfall thresholds (Matelloni et al., 2011; Gruzzetti et al., 2007; Aleotti, 2004; Wilson, 1997).*

*Each of these LHZ techniques has its own advantage and disadvantage owing to certain uncertainties on account of factors considered or methods by which factor data are derived (Carrara et al.,1995).*

*"Limit equilibrium types of analyses for assessing the stability of earth slopes have been in use in geotechnical engineering for many decades. The idea of discretizing a potential sliding mass into vertical slices was introduced in the 20th century. During the next few decades, Fellenius introduced the Ordinary method of slices (Fellenius, 1936) . In the mid1950s Janbu and Bishop developed advances in the method (Janbu, 1954; Bishop, 1955). The advent of electronic computers in the 1960's made it possible to more readily handle the iterative procedures inherent in the method, which led to mathematically more rigorous formulations such*

*as those developed by Morgenstern and Price and by Spencer (Morgenstern and Price, 1965; Spencer, 1967)."*

(1) There are plenty of free software (see for example TRIGRS model of USGS). You should cite them too and specify the differences with your model. Line 36-38.

(2) To address this question the following text (including new references) has been added into the introduction section:

(3) *"Limit equilibrium types of analyses for assessing the stability of earth slopes have been in use in geotechnical engineering for las year. Currently, the vast majority of stability analyses using **this method of equilibrium limit** are performed with commercial software like SLIDE V5, SLOPE/W, Phase2, GEO-Slope, GALENA, GSTABL7, GEO5 and GeoStudio, entre otros [Mousavi, 2017; Acharya et al., 2016a; Acharya et al., 2016b; Jiao et al., 2013; Gonzalez de Vallejo et al., 2002). Other models of slope stability based on the theory of limit equilibrium are still being studied, as is the case of the SSAP model (Borselli, 2016), but in this case a General equilibrium method model is applied."*

*"There are other types of software based on the modeling of the probability of occurrence of shallow landslides LHZ, in more extensive areas using GIS technology and MDE, as is the case of deterministic software TRIGRS ,SINMAP, SHALSTAB, GEOtop/GEO-FS, R-Slope.stability among others (Tran et al., 2018; Alvioli and Baum, 2016; Reid et al., 2015; Mergili et al., 2014a; Mergili et al., 2014b; Mergili et al., 2014c; Baum, 2008; Simoni et al., 2008; Rigon et al., 2006; Pack, 2001). They are widely used models for calculating the time and location of the occurrence of shallow landslides caused by rainfall at the territorial level; some even in three dimensions, in order to obtain a probabilistic interpretation of the factor of safety.*
*Currently other approaches / theoretical studies for landslide prediction are used (for triggering and / or propagation) (Matelloni et al., 2017; Martelloni and Bagnoli, 2014).*

*The idea of discretizing through this tool proposed (TS), the potential slip mass in the critical profile of the slope, once we have detected through the HZD programs unstable areas, is one of the achievements of this model. This calculation tool is not limited to shallow landslides and debris flows, but allows analysis of deep and rotational landslides, which others do not allow. Using the infiltration factor of Spencer $r_u$ we introduce the*

*hydrological variable by infiltration to the stability calculation of the slope."*

**(1) Please, clarify this sentence (line 41-42).**

(2) We rewrite the sentence, with another clarifying development

(3) *"Second, sometimes in these commercial software, the introduction of the parameters to perform the calculations, are not very interactive."*

**(1) Add citations in support of this sentence (line 48-49).**

(2) Reference is added

(3) *"These methods allow us to analyse almost all types of landslides, such us translational, rotational, topple, creep and fall, among others (Zhou and Cheng, 2013; Wan et al, 2016).*

**(1) Make the acronym explicit the first time it is introduced (line 58)**

(2) FOS, is the safety factor, it is simplified and denominated throughout the text as Fs (safety of factor).

(3) *"Software such as the programmes mentioned above provide useful tools for determining the stability through the $F_S$ (safety of factor)….."*

**(1) Are these software free or commercial? It would be better to add them also in the first paragraph of the introduction, among others (File 60).**

(2) It is a commercial calculation software on Slide V5 and the STB 2010 is free. The suggested change is done and the reference is entered on line 49 of the introduction. I introduce a new free software. Is removed from line 60.

(3) *"For the stability analysis, different approaches can be used, such as the limit equilibrium methods [Cheng et al., 2007; Liu et al., 2015], the finite elements method [Griffiths et al., 2007; Tschuchnigg et al., 2015; Griffiths, 2015] and the dynamic method [Jia et al., 2008], among others (Slide V5, STB 2010 and SSAP 2018)"*

**(1) I believe this is not true, Please, verify this statement with more literature review. (line 62-64)**

(2) The sentence has been deleted.

**(1) For consistency I suggest referring to the factor of safety as FOS or Fs. (line 75)**

(2) The suggestion is accepted and Fs is taken as the factor of safety and we remove from the text the term FOS.

(3) *"The primary result of this model was a stability index, namely the minimum Fs,........"*

(1) **Please, explicit in the text the meaning of R and alpha. (Line 98).**

(2) The text now is as follows:

*(3) "In the equation 3, R is the radius of the curvature and α is the angle of the slope referred to each slice"*

(1) **Please, describe also the meaning of alpha, b and h.**

(2) The suggestion is done.

*(3) "α is the angle of the slope referred to each slice, b is the slice width and h is the mean height of slice (if the height is not constant)."*

**(1) Make the acronym explicit the first time it is introduced in the text, remove (lines 120-121).**

(2) TS, is the proposed model (terrain stability), but I think it is convenient to delete the line as the reviewer indicates.
**(3) Remove:** *Spencer's method [Spencer, 1967] is more precise and simple in the TS model.*

**Add reference. (line 122).**
(1) It is done as follows:
(2) *"Taking into account these elements, the Fs is then obtained from the following expression (Spencer, 1967)."*

**(1)The terms of this equation must be explicited. (line 123)**
(2) It is accepted.
(3) *"Where $\phi'$ is the friction angle at the fracture surface, **u** is the pore pressure at the fracture zone, c´ is the soil cohesion, $\alpha$ is the angle at the base of the slice, W is the external vertical forces and b the width of the slice.*

**(1)Please, check the use of talus here. Do not use it as a synonym of slope.**
(2) It is accepted.
(3) *"As mentioned, the minimum Fs to consider a slope stable is equal to 1."*

**(1)Why is the pore pressure mentioned just now?**
(2) I have made a change at the proposal of the reviewer 3 and in view of its indication.

**(3) Enter on line 155:**

*"When solving the normal and parallel forces at the base of the slice of the five acting forces, we obtain (Q), resulting from the forces between slices:*

$$Q = \frac{\frac{c'b}{F}\sec\alpha + \frac{\tan\phi'}{F}(W\cos\alpha - ub\sec\alpha) - W\sin\alpha}{\cos(\alpha - \theta)[1 + \frac{\tan\phi'}{F}\tan(\alpha - \theta)]}$$

*In this expression, u is the pore pressure (permanent interstitial pressure) at the base of the slice and the weight of the slice is determined by W. If we assume that the soil is uniform and its density ($\gamma$) also, the weight of a slice of height h and width b can be written:*

$$W = \gamma bh \qquad\qquad "$$

**Enter on line 164-172:**

*The factor $r_u$ is a coefficient of pore pressure (interstitial pressure coefficient), which determines the rain infiltration factor on the slopes. As it is well known, the water that infiltrates the soil may produce a*

*modification of the pore pressure, affecting its resistant capacity. This factor may vary from 0 (dry conditions) to 0.5 (saturated conditions). In the article of Spencer (Spencer, 1967), assuming a homogeneous pore-pressure distribution as proposed by Bishop and Morgenstern (1960), the mean pore-pressure on the base of the slice can be written like the equation 7."*

**(1)There is the need to explain and show in a map where this test has been carried out.  Is this a private company? (Lines 181-183).**

(2) The following sentence is deleted: *"According to the in situ test carried out by the Geoner SL laboratory, the soil is a silty clay from Gibraltar Flish"*

The Flysch of Gibraltar is not the soil analyzed later in the case study in Viñuela, it is only used in this section of the document to see how the proposed program works. It is replaced by:

(3)*"Geotechnical data of a cohesive soil of the Flysch type of Gibraltar, (Vallejo et al., 2002)"*.

**(1) If possible, show it in figure 2. (Line 194-195). The user?  (line 201). Is this the same point of line 194? (line 207)**

(2) The point (xc, yc) is entered in the output graph of the program; but for a better understanding of the code, we introduce a new figure in the manuscript, only with the initial curve.

The function that is minimized with the proposed code is the safety factor Fs calculated with the Spencer method and subject to restrictions on the lower cut point with the slope (0, yt) as well as on the position of the center of the turn of the critical curve (xc, yc). Given an initial curve (yellow curve) characterized by the point x = (xc, yc, yt), the fmincon function of Matlab is used in our code to obtain the critical point (xc *, yc *, yt *) so the code draws the critical curve (green curve), where the safety factor is minimal. The following paragraph and figure are entered in line 183.

There is a typo, it is (xc, yc) and not (x0, y0) in the line 207.

The following sentence is deleted: *"According to the in situ test carried out by the Geolen SL laboratory, the soil is a silty clay from Gibraltar Flish"*

The Flysch of Gibraltar is not the soil analyzed later in the case study in Viñuela, it is only used in this section of the document to see how the proposed program works. It is replaced by:

(3) *"Figure 2 shows the results of applying the Terrain Stability model to an irregular slope, including the initial and final points of the first failure circle (shown in yellow). This circle corresponds with the initial value introduced by the user into the FSOLVE function. The points of the slope (topographic) are extracted from a DEM model in ArcGIS 10 (Glennon et al., 2008). The slope height is equal to 15 m, and the soil is uniform with the following nominal properties: $\gamma$ = 19500 N/m³, $\phi$ = 22º, c = 15000 N/m², u = 0 N/m². "*

[Figure]

***Figure 2***. *In this example, the center coordinates are equal to xc = 7 m; yc = 14 m, and the lower cut with the slope coordinates (P1 point) equal to xt = 0 m, and t = 0 m, data that the user introduces.*

*The code works as follows: the initial circular failure curve is plotted using the FPLOT tool, as shown in Figure 2 (yellow line). In this example, the centre coordinates are equal to xc = 7 m; yc = 14 m and the lower cut with the slope coordinates (P1 point) equal to xt = 0 m, yt = 0 m. The Fs obtained was 1.6, which is, in principle, a stable slope. It must be taken into account that the mass susceptible*

*to slipping must be divided into N pieces equal to the number of slices; in this example, the mass was divided into N = 500 slices, the value of N is entered into the user code, plus divisions of the sliding mass, more accuracy but greater need for computer capacity.*

**(1) This rectangular box should be shown in Figure 2. Too many coordinates. Also, if possible, show it in figure 2. (lines 213-215) (line 220)**

(2) In figure 2, we could not draw the center of critical coordinates, since it indicates the calculation un $y_c$= 28,1091 m. and the scale of Y(m) would in this case reach 16 m., but if necessary, it could be drawn by changing the scale of the drawing. In any case, as indicated by the initial point in Figure 2, the possibility is shown.

(3) *"Figure 2, would be renamed figure 3.*

[Figure]

**(1) How this parameter have been calculated? (line 236) How this parameter have been calculated? (line 398).**

(2) The pore pressure will be hydrostatic, defined by: $u = \gamma_w(\mathrm{h} - \mathrm{h}_w)$, $\gamma_w$ is the saturated density of soil, h and $\mathrm{h}_w$ is the difference between saturated and dry height.

$$r_u = \frac{u}{\gamma h}$$

In this expression, u is the pore pressure (permanent interstitial pressure) at the base of the slice, γ is the density of soil, h is the mean height of slice (if the height is not constant) and the weight of it affects the W evaluation.

(3) *"The pore pressure will be hydrostatic, defined by: $u = \gamma_w(h - h_w)$, $\gamma_w$ is the saturated density of soil, h and $h_w$ is the difference between saturated and dry height. The calculation of the infiltration factor is calculated with the following equation:*

$$r_u = \frac{u}{\gamma h}$$

*The factor $r_u$ is a coefficient of pore pressure (interstitial pressure coefficient), which determines the rain infiltration factor on the slopes. As it is well known, the water that infiltrates the soil may produce a modification of the pore pressure, affecting its resistant capacity. This factor may vary from 0 (dry conditions) to 0.5 (saturated conditions). In the article of Spencer (Spencer, 1967), assuming a homogeneous pore pressure distribution as proposed by Bishop and Morgenstern (1960), the mean pore-pressure on the base of the slice can be written like the equation 7."*

**(1)Please, resize the Figure 2 and 3 in order to be comparable.**
(2) Scales are equalized.

(3)

[Figure]

[Figure]

**(1)It would be good to put all these information in a GIS map**
(2)We introduce map GIS:
(3)*"In this case we have looked for the map in the IGN, National GeographicInstitute:websitehttp://centrodedescargas.cnig.es/Centro Descargas/index.jsp, We have downloaded the raster map MTN25, which is a 1: 25.000 topographic map, with ETRS 89 coordinates, UTM projection. It is a File generated by means of a digital rasterization (vector to raster conversion) georeferenced, specifically*

*we have downloaded sheet number 1039, which is the one corresponding to the landslide zone.*

*Once we have downloaded the ecw file, we open it with any GIS software, be it the ArcGis, the Land basic Map, among others. With this map we can have the topographic map and make the necessary profiles for the study and analysis of the landslide, normally the most unfavourable profile of the topography is studied in this case."*

[Figure]

[Figure]

**(1)Add reference. (lines 259-261)**

(2)The references are introduced as follows

(3)*"This type of mechanism is characteristic of homogeneous cohesive soils, as was the one analysed here (Cornforth, 2005; Rahardjo et al., 2007; Lu and Godt, 2008)."*

**(1)You did not mentioned the date of the failure, therefore the rainfall series does not make sense here.(line 319-320). Highlight the date of the landslide in this figure. Label the axes and show the months in the x-axis. How far is from the site? Where you can find the data?**

(2)The landslide analyzed began in February 2010, ending in March of that same year, hence we have indicated those specific months in the histogram. Hence, we believe that the rain histogram supplied by the Meteorological Agency of Spain in that area is necessary, through the Viñuela Weather Station.

(3)*"The landslide analyzed began in February 2010, ending in March of that same year.*

*Figure 7. Rainfall histogram at La Viñuela from August 2009 to April 2010. The data to make the rain histogram, has been supplied by the Meteorological Agency of Spain, through the Meteorological Station of Viñuela."*

**(1)The units of measurements should be consistent with the figure. Line 332.**
(2)A syntax error is detected.

(3)*"It can be observed that large amounts of precipitation fell during the months of December, January, February and March of 2010, with peaks of most 60 l/m2 in a single day (January and February). In total, 890 l/m2 fell in the 2009-2010 hydro cycle, which ended at the end of April 2010."*

**(1)Please, clarify this sentence**
(2)A syntax error is detected

(3)*"We applied the TS model using topographic data obtained from the ArcGIS 10 software program. We did so to obtain the degree of*

*stability of the sliding land based on the angle of internal friction, the cohesion, the density and the angle of the slope we analyzed. Figure 9 shows the analytical results from the real slope, by studying and analyzing the most unfavorable profile of the landslide studied. In addition we compared the results given by the developed TS model and the results given by STB 2010 model, using free surfaces in both cases. In our model the worst curve (shown in green) was calculated automatically from the initial curve (show in blue), resulting in $F_S$ = 2.300, in the dry state."*

**(1)See comments in the introduction. (Line 415)**
(2)Clarify the sentence
(3)*"our original algorithm TS model appears to be more efficient and accurate."*

**(1) This list is disjuncted from the rest of the text and probably belongs to the chapter 2 (description of the model). (Lines 415-444).**
(2) The change to the description of the model in section 2 has been done and can be checked in previous comments.

---

## Author Comment (AC2) · 10 Jan 2019

We maintain that the manuscript is well within the aims of NHESS. We provide in the attached supplement, the constestations and contributions to the questions raised by you.

Thanks for your comments

regard greeting

GENERAL COMEMTS.

(1) The manuscript describes a numerical approach to slope stability, and the corre-

sponding original software. The model is two-dimensional, and its applicability is limited to a single slope; advantages are the software being freely available and inclusion of wet soil conditions, apparently missing in existing commercial software.

I believe that the manuscript suffer from several limitations, and in my opinion is not suitable for publication in NHESS. I will try and motivate my opinion in three different sections, as requested by NHESS reviewing guidelines.

(2) We maintain that the manuscript is well within the aims of NHESS. Undoubtedly, several papers on rainfall thresholds and landslides induced by intense rainfall events. But our novelty lies in the development of an original code with programming in Matlab, with the ability to predict well the slip failure of the curve and the area of the surface, taking into account the rain infiltration factor $r_u$ of the Spencer method.

We will take into consideration some of the assessments that we believe will improve the document. We have used a large number of the bibliographical references provided. Changes that we mention below.

The review say: "The model is two-dimensional, and its applicability is limited to a single slope. . . . . . . . ..."

We cannot agree with this sentence because in the proposed example we have taken the topographic profile of the critical analyzed slope, although we can analyze all the profiles that we want of slope and landslide that we need with our code. The coordinates of the profile have been obtained from a topographic map of the slope and we have obtained it through a raster map and a GIS application.

The advantage over other 3D models and similar, is that this proposed code deals with the ability to predict a landslide failure curve and the slope factor of safety with a terrain stability (TS) analysis. Has the ability to well-predict the landslide shape and area.

(1) I believe that the material in the manuscript is organized in a rather confusing way, and that key sections of the text do not contain the information they are supposed to.

The Title suggests that the paper deals with landslide "risk", while it describes a numerical model for slope stability assessment. The generally accepted definition of "risk" associated with a natural hazard is the product, or the combination, of the likelihood of an event of the given kind ("hazard") and "exposure", or "vulnerability", of human life and infrastructure to that kind of hazard. Moreover, the generally accepted definition of "hazard" is, in turn, the product of spatial probability, temporal probability and magnitude of an event of the given type to occur. The model described in the manuscript deals with spatial and magnitude assessment of landslides; it is not clear to me whether a temporal component is included.

(2) Our code does not include the temporary component that indicates us, so we understand that we should better adjust the title to the proposed code:

(3) "Development and validation of the Terrain Stability model for assessing landslide instability during heavy rain infiltration."

(1) Surely we cannot speak about "probability" here, because the model obtains a factor of safety, which is clearly NOT a probability. In order to obtain a probabilistic interpretation of the factor of safety, one needs to perform additional, non trivial steps. See, for example:.....

Moreover, it is not true that the model itself includes an assessment of vulnerability, which must be taken into account separately and, most importantly, with additional (and often difficult to obtain) data. The Title also mention validation of the model, which was actually performed in a rather qualitative way. It also mentions the expression:

"during heavy rain infiltration", which is not actually substantiated in the manuscript since, again, no explicit time dependence is implemented as the word "during" would suggest, and no actual "infiltration" is considered, but only its effective result - namely, an effective value for pore pressure calculated at an arbitrary depth under the soil surface. At least, this is what I can understand after reading the whole manuscript. I will give more details below.

The Abstract contains unnecessary information (the first two, long sentences), a few inaccuracies (see below) and, most importantly, fails to properly and succinctly introduce the methods, results and conclusions obtained in the manuscript. A GIS support is mentioned, while the whole code is implemented in Matlab.

(2) We disagree, a safety factor to the stability of a slope determines a number by which it indicates if there is a probability or not of being stable, if Fs> 1, it will be stable and there will be a great probability of stability, if it is less than 1, it will be unstable and there will be a likelihood of landslide on the slope.

Regarding the indicated references, we take it to include them in the introduction of our manuscript.

The rainwater infiltration is justified by a histogram figure 7 of our manuscript. The depth of calculation of the pore pressure is not arbitrary since our code calculates the surface and the critical sliding curve, so that the height of the slices in which We divide the land mass susceptible to sliding on the slope does not It is arbitrary with our code.

We have to say that the GIS support is only used to obtain the topographic coordinates of the critical profile to study in our code; hence it is only implemented in Matlab. But the support of the GIS system is there. There are other programs that have GIS support, but they have other different characteristics and are mostly used to analyze areas of slip instability normally shallow at the territorial level, using other models such as the slope limit. In our model we have the two options of stability calculation on slope, on the one hand we have translational or shallow landslides and on the other hand we can do stability calculation for rotational and deep landslides.

The versatility of this code lies in its engineering resolution, it is not a development in basic science but it is a very useful tool in engineering resolution, which in our opinion is a perfectly legitimate field for an NHESS publication. Our code has originality in front of the indicated commercial software, besides not being of payment, to raise a model of calculation with restrictions that the user imposes, by means of the Fmincon function

of Matlab, in addition to the incorporation of a pore pressure factor by means of the application of the ru factor of the Spencer method. It is proposed in light of having raised other reviews, a more explicit explanation and development of the problem of this natural hazard and existing software's.

(1) Sections 2 and 3, devoted to a description of the methodology implemented in the model, are confusing, and I cannot understand what are the assumptions and the relevant details of the method implemented in the software, and whether it is a novel enough approach. I will give more details later on. Section 4.3 is devoted to the description of the results obtained using the proposed model. This section is very confusing, again. I believe that the comparison of the results of the proposed model with another model, and with a real landslide scenario, are presented in an unsatisfactory way, since they are qualitative almost everywhere and it is difficult to understand what the quantitative comparisons refer to. Moreover, there is a large fraction of text which does not pertain to results but to the methodology itself.

Eventually, in Section 5, devoted to describe conclusions of the manuscript, again I do not find enough evidence of actual conclusions drawn from the results. In addition to repeating already mentioned concepts in a, in my opinion, misleading way (i.e., use of "prediction", of "time", etc.), there are a couple of expressions which, I believe, are not allowed in assessing the conclusions in a scientific paper. First, the Authors state that the proposed model "defines fairly well areas that intuitively appear to be susceptible to landslides and defined rigorously the failure curve". In this sentence, "fairly well" and "intuitively" are not good enough to assess the predicting performance of a quantitative model. Moreover, the "rigorous" definition of slip surfaces does not appear to be substantiated by the presented results, as I will explain at length in the following. Then, the expression "this model is probably the most powerful tool for determining slope stability", is again not substantiated by the presented results. Eventually, a reference to the SINMAP model comes out of the blue, in the second-last line, which is unjustified.

(2) About the comments made in this section, in general we cannot agree with these

comments, by following:

The review, recommending the rejection of the manuscript, gives us to think that the reviewer has not understood the objective and the code developed, and explained within the manuscript. Probably by our fault, bus we think that all the changes conducted has improved the document a lot.

The reviewer criticized our local code and compare it with regional codes such as TRIGRS, SINMAP, SHALSTAB (based on GIS) among others, which are totally different codes, with totally different objectives and based on totally different fundamental ideas.

In any case, in the revision of the paper we intend to introduce in the introduction section at the request of the other reviewers the reference to these stability programs at territorial level. This software's are based on raster maps, with a resolution limited by their pixel quality, and with probabilistic calculations established by a previous slope catalogue. However, in our case, we analyze the field extracted data for a determined slope. Also our code allows us to use the number of slices that the user wants, on the contrary, the reviewer state that this is an aleatory thing.

This is a good feature allowing the user to adapt their calculations to their necessities in terms of precision and computational time. In addition, he/she suggested that the Spencer resolution is in some way random, and we believe that this does not deserve any additional comment from our side. The code that we have developed can be used in civil engineering to study the slope stability, with the capacity to predict the superficial landslides and deeper ones, with the addition of water infiltration, as we have been used in the UME (Military Emergencies Unit).

Also the reviewer asks for the MDE mode, while we only used it to extract the 2D topographical profile. These data, as referenced in the manuscript, can be obtained from the Geographical National Institute, You can download it on your website:

http://centrodedescargas.cnig.es/CentroDescargas/index.jsp

He also ask for the hydrological data which has been described in the manuscript and in the referenced bibliography in a previous paper published by the authors. The geotechnical and lithological data of the analyzed slope, say that they are random and difficult to find; this is not true, in our case study and analysis of our code, if geotechnical tests have been performed (table 1, table 2 and table 3 of the manuscript) to be able to demonstrate the calibration of our code, which, by the way, the critical curve coincides the reality of the real landslide occurred, is shown in the profile analyzed (figure 6 of the manuscript), a fact that could not be achieved with the stability programs probabilistic that you recommend.

In any case, the data that you say is difficult to obtain, it is not true, because you can obtain them from the Mining Geological Institute in this case from Spain. I attach the email address: http://www.igme.es/

In this page we have in the download area of raster maps where lithology by delimited leaves appears; that is, we can obtain the topographic profile and in the case of not having geotechnical tests, we can extract them from these raster maps that delimit the lithology. Once the lithology is delimited, we can obtain its geotechnical characteristics in existing tables in special geological engineering bibliography, among others:

González de Vallejo, L., Ferrer, M., Ortuno, L., and Oteo, C. (2002). Geological Engineering. Madrid: Prentice Hall.

Please also note the supplement to this comment:
https://www.nat-hazards-earth-syst-sci-discuss.net/nhess-2018-192/nhess-2018-192-AC2-supplement.pdf

**Supplement:**

Comments by Editor:
But the Editor will have the final decision on that.

Dear Authors,

You - as the contact author - are requested to individually respond to all referee comments (RCs) by posting final author comments on behalf of all co-authors no later than 13 Jan 2019 (final response phase) at: https://editor.copernicus.org/nhess-2018-192/final-response.

**Comments by Anonymous Referee #2 (nhess-2018-192-RC2)**

[Answers in blue]

**GENERAL COMEMTS.**

(1) The manuscript describes a numerical approach to slope stability, and the corresponding original software. The model is two-dimensional, and its applicability is limited to a single slope; advantages are the software being freely available and inclusion of wet soil conditions, apparently missing in existing commercial software.

I believe that the manuscript suffer from several limitations, and in my opinion is not suitable for publication in NHESS. I will try and motivate my opinion in three different sections, as requested by NHESS reviewing guidelines.

(2) We maintain that the manuscript is well within the aims of NHESS. Undoubtedly, several papers on rainfall thresholds and landslides induced by intense rainfall events. But our novelty lies in the development of an original code with programming in Matlab, with the ability to predict well the slip failure of the curve and the area of the surface, taking into account the rain infiltration factor ru of the Spencer method.

We will take into consideration some of the assessments that we believe will improve the document. We have used a large number of the bibliographical references provided. Changes that we mention below.

**The review say:** *"The model is two-dimensional, and its applicability is limited to a single slope……….."*

We cannot agree with this sentence because in the proposed example we have taken the topographic profile of the critical analyzed slope, although we can analyze all the profiles that we want of slope and landslide that we need with our code. The coordinates of the profile have been obtained from a topographic map of the slope and we have obtained it through a raster map and a GIS application.

The advantage over other 3D models and similar, is that this proposed code deals with the ability to predict a landslide failure curve and the slope factor of safety with a terrain stability (TS) analysis. Has the ability to well-predict the landslide shape and area.

(1) I believe that the material in the manuscript is organized in a rather confusing way, and that key sections of the text do not contain the information they are supposed to.

The Title suggests that the paper deals with landslide "risk", while it describes a numerical model for slope stability assessment. The generally accepted definition of "risk" associated with a natural hazard is the product, or the combination, of the likelihood of an event of the given kind ("hazard") and "exposure", or "vulnerability", of human life and infrastructure to that kind of hazard. Moreover, the generally accepted definition of "hazard" is, in turn, the product of spatial probability, temporal probability and magnitude of an event of the given type to occur. The model described in the manuscript deals with spatial and magnitude assessment of landslides; it is not clear to me whether a temporal component is included.

(2) Our code does not include the temporary component that indicates us, so we understand that we should better adjust the title to the proposed code:

(3) *"Development and validation of the Terrain Stability model for assessing landslide instability during heavy rain infiltration."*

(1) Surely we cannot speak about "probability" here, because the model obtains a factor of safety, which is clearly NOT a probability. In

order to obtain a probabilistic interpretation of the factor of safety, one needs to perform additional, non trivial steps. See, for example:.....

Moreover, it is not true that the model itself includes an assessment of vulnerability, which must be taken into account separately and, most importantly, with additional (and often difficult to obtain) data. The Title also mention validation of the model, which was actually performed in a rather qualitative way. It also mentions the expression:

"during heavy rain infiltration", which is not actually substantiated in the manuscript since, again, no explicit time dependence is implemented as the word "during" would suggest, and no actual "infiltration" is considered, but only its effective result - namely, an effective value for pore pressure calculated at an arbitrary depth under the soil surface. At least, this is what I can understand after reading the whole manuscript. I will give more details below.

The Abstract contains unnecessary information (the first two, long sentences), a few inaccuracies (see below) and, most importantly, fails to properly and succinctly introduce the methods, results and conclusions obtained in the manuscript. A GIS support is mentioned, while the whole code is implemented in Matlab.

(2) We disagree, a safety factor to the stability of a slope determines a number by which it indicates if there is a probability or not of being stable, if Fs> 1, it will be stable and there will be a great probability of stability, if it is less than 1, it will be unstable and there will be a likelihood of landslide on the slope.

Regarding the indicated references, we take it to include them in the introduction of our manuscript.

The rainwater infiltration is justified by a histogram figure 7 of our manuscript. The depth of calculation of the pore pressure is not arbitrary since our code calculates the surface and the critical sliding curve, so that the height of the slices in which We divide the land mass susceptible to sliding on the slope does not It is arbitrary with our code.

We have to say that the GIS support is only used to obtain the topographic coordinates of the critical profile to study in our code; hence it is only implemented in Matlab. But the support of the GIS system is there. There are other programs that have GIS support,

but they have other different characteristics and are mostly used to analyze areas of slip instability normally shallow at the territorial level, using other models such as the slope limit. In our model we have the two options of stability calculation on slope, on the one hand we have translational or shallow landslides and on the other hand we can do stability calculation for rotational and deep landslides.

The versatility of this code lies in its engineering resolution, it is not a development in basic science but it is a very useful tool in engineering resolution, which in our opinion is a perfectly legitimate field for an NHESS publication. Our code has originality in front of the indicated commercial software, besides not being of payment, to raise a model of calculation with restrictions that the user imposes, by means of the Fmincon function of Matlab, in addition to the incorporation of a pore pressure factor by means of the application of the $r_u$ factor of the Spencer method. It is proposed in light of having raised other reviews, a more explicit explanation and development of the problem of this natural hazard and existing software's.

(1) Sections 2 and 3, devoted to a description of the methodology implemented in the model, are confusing, and I cannot understand what are the assumptions and the relevant details of the method implemented in the software, and whether it is a novel enough approach. I will give more details later on. Section 4.3 is devoted to the description of the results obtained using the proposed model. This section is very confusing, again. I believe that the comparison of the results of the proposed model with another model, and with a real landslide scenario, are presented in an unsatisfactory way, since they are qualitative almost everywhere and it is difficult to understand what the quantitative comparisons refer to. Moreover, there is a large fraction of text which does not pertain to results but to the methodology itself.

Eventually, in Section 5, devoted to describe conclusions of the manuscript, again I do not find enough evidence of actual conclusions drawn from the results. In addition to repeating already mentioned concepts in a, in my opinion, misleading way (i.e., use of "prediction", of "time", etc.), there are a couple of expressions which, I believe, are not allowed in assessing the conclusions in a scientific paper. First, the Authors state that the proposed model "defines fairly well areas that intuitively appear to be susceptible to landslides and defined rigorously the failure curve". In this sentence, "fairly well" and

"intuitively" are not good enough to assess the predicting performance of a quantitative model. Moreover, the "rigorous" definition of slip surfaces does not appear to be substantiated by the presented results, as I will explain at length in the following. Then, the expression "this model is probably the most powerful tool for determining slope stability", is again not substantiated by the presented results. Eventually, a reference to the SINMAP model comes out of the blue, in the second-last line, which is unjustified.

(2) About the comments made in this section, in general we cannot agree with these comments, by following:

The review, recommending the rejection of the manuscript, gives us to think that the reviewer has not understood the objective and the code developed, and explained within the manuscript. Probably by our fault, bus we think that all the changes conducted has improved the document a lot.

The reviewer criticized our local code and compare it with regional codes such as TRIGRS, SINMAP, SHALSTAB (based on GIS) among others, which are totally different codes, with totally different objectives and based on totally different fundamental ideas.

In any case, in the revision of the paper we intend to introduce in the introduction section at the request of the other reviewers the reference to these stability programs at territorial level. This software's are based on raster maps, with a resolution limited by their pixel quality, and with probabilistic calculations established by a previous slope catalogue. However, in our case, we analyze the field extracted data for a determined slope. Also our code allows us to use the number of slices that the user wants, on the contrary, the reviewer state that this is an aleatory thing.

This is a good feature allowing the user to adapt their calculations to their necessities in terms of precision and computational time. In addition, he/she suggested that the Spencer resolution is in some way random, and we believe that this does not deserve any additional comment from our side. The code that we have developed can be used in civil engineering to study the slope stability, with the capacity to predict the superficial landslides and deeper ones, with the addition of water infiltration, as we have been used in the UME (Military Emergencies Unit).

Also the reviewer asks for the MDE mode, while we only used it to extract the 2D topographical profile. These data, as referenced in the manuscript, can be obtained from the Geographical National Institute, You can download it on your website:

http://centrodedescargas.cnig.es/CentroDescargas/index.jsp

He also ask for the hydrological data which has been described in the manuscript and in the referenced bibliography in a previous paper published by the authors. The geotechnical and lithological data of the analyzed slope, say that they are random and difficult to find; this is not true, in our case study and analysis of our code, if geotechnical tests have been performed (table 1, table 2 and table 3 of the manuscript) to be able to demonstrate the calibration of our code, which, by the way, the critical curve coincides the reality of the real landslide occurred, is shown in the profile analyzed (figure 6 of the manuscript), a fact that could not be achieved with the stability programs probabilistic that you recommend.

In any case, the data that you say is difficult to obtain, it is not true, because you can obtain them from the Mining Geological Institute in this case from Spain. I attach the email address: http://www.igme.es/

In this page we have in the download area of raster maps where lithology by delimited leaves appears; that is, we can obtain the topographic profile and in the case of not having geotechnical tests, we can extract them from these raster maps that delimit the lithology. Once the lithology is delimited, we can obtain its geotechnical characteristics in existing tables in special geological engineering bibliography, among others:

González de Vallejo, L., Ferrer, M., Ortuno, L., and Oteo, C. (2002). Geological Engineering. Madrid: Prentice Hall.

**SPECIFIC COMMENTS**

(1) In the Abstract, in addition to unrelevant information in the first two sentences already mentioned above, I believe that a few other ambiguities exist. It is stated that "Climate is one of the main factors [affecting slope stability, Ed.], especially when large amounts of rainwater are absorbed in short periods of time". The paper does not discuss climate effects on landslides, or correlations between the

different factors determining the climate of a given region and landslides. Thus, this should not appear in the Abstract, which must contain a short description of the specific topic discussed in the paper; maybe in the introduction, if a sufficiently clear link is made with the topic of the paper.
A quantitative relationship with climate (actually, climate change), rainfall events and slope stability including actual time-dependent account for rainfall infiltration can be found in, e.g....

(2) In the abstract say:

*"The geological stability of slopes is affected by several factors, such as climate, earthquakes, lithology and rock structures, among others.........."*

That is to say in the abstract we talk about all the factors that affect the stability of the slopes:

Lithology, earthquakes, types of rocks and also the climate and within the climate we particularize it to the rainfall and in particular to the water infiltration.
We go from more to less, in the proposed code or algorithm we develop a slope optimizer with the possibility of integrating it with a hydrological factor $r_u$.

This coefficient was obtained with the following expression:

$$r_u = \frac{u}{\gamma h}$$

The parameter u is the interstitial pressure at the base of the segment calculation; assuming a homogeneous distribution of the pressure (as other authors suggested such as Spencer, Bishop and Morgenstern).

This model is especially useful for predicting the risk of landslides in scenarios of heavy unpredictable rainfall. A hydrological steady-state assumption was incorporated into this approach. The model, called Terrain Stability (TS), was developed and programmed in MATLAB.

We do not understand the lack of consideration of our investigation in view of the indicated. But when we are in a magazine that

encourages scientific debate and the participation of researchers, com has been positive in this case due to the large number of entries to this manuscript.

Then it includes a series of references that we will include if the editor estimates us the publication of this manuscript, these references will be included in the introduction as existing models in a later revision; but that has nothing to do with our code, are probabilistic models and territorial level for the study of landslide hazard maps. Our code exhaustively studies the critical curve of landslide, something that the others that it indicates do not do to us.

(1) The Authors claim that the model is "supported by a GIS", which is not true. The only step in which a GIS can (can) be used is when they mention that the terrain profile was obtained fom a DEM. The model is coded in Matlab, and not within a GIS. The statement "the model is especially useful for predicting ... scenarios of heavy unpredictable rainfall" is very bold. How can account for rainfall into a mathematical model, if the rainfall is unknown?!?

(2) We have already answered this same circumstance, which was already revealed in the General Comments.

(1) In the Introduction, in addition to what I have written above. At lines 35-36, the Authors refer to a 2002 paper commenting "nowadays"; I believe that more recent papers exist, other than a 16-years-old one. FOS is used but not defined. Moreover, throughout the Manuscript, the Authors refer to factor of safety using both "FOS" and "Fs", apparently for no good reason.

(2) We accept the suggestion and add new more current references, as we will show below in the section (3).

We also decided, as other reviewers have indicated, to use only one denomination to define the safety factor. In this case we have chosen the denomination Fs, which is the one that uses the code as a safety factor; we will eliminate the FOS denomination from the manuscript.

(3) "*Limit equilibrium types of analyses for assessing the stability of earth slopes have been in use in geotechnical engineering for las year. Currently, the vast majority of stability analyses **using this***

*method of equilibrium limit* *are performed with commercial software like SLIDE V5, SLOPE/W, Phase2, GEO-Slope, GALENA, GSTABL7, GEO5 and GeoStudio, among others [Mousavi, 2017; Acharya et al., 2016a; Acharya et al., 2016b; Jiao et al., 2013; Gonzalez de Vallejo et al., 2002). Other models of slope stability based on the theory of limit equilibrium are still being studied, as is the case of the SSAP model (Borselli, 2016), but in this case a General equilibrium method model is applies."*

(1) Section 2, where details of the models are described, is rather confusing to me. After the relevant equations are introduced (missing the definition of a few quantities here and there), it is stated that these coupled equations must be solved to obtain the factor of safety Fs and the angle theta. Then, theta is assumed as constant, for no apparent reason, other than the sentence that "it provides optimal results", with no further explanation or justification. The whole meaning of theta should be explained in a better way, in my opinion. Then, the role of pore pressure is introduced. I believe it is impossible to understand why they Authors refer to a "pore pressure distribution" and then use a single value (effective coefficient?) for it, or if they actually use a distribution. Moreover, it is impossible to understand if a time dependence - according to actual rainfall infiltration as a function of, well, rainfall intensity and varying water content in the soil, was taken into account or not.

(2) We have made a change at the proposal of the reviewer 3 and reviewer 2, and in view of its indication.

(3) **Enter on line 155:**

*"When solving the normal and parallel forces at the base of the slice of the five acting forces, we obtain (Q), resulting from the forces between slices:*

$$Q = \frac{\dfrac{c'b}{F}\sec\alpha + \dfrac{\tan\phi'}{F}(W\cos\alpha - ub\sec\alpha) - W\sin\alpha}{\cos(\alpha - \theta)\left[1 + \dfrac{\tan\phi'}{F}\tan(\alpha - \theta)\right]}$$

*In this expression, u is the pore pressure (permanent interstitial pressure) at the base of the slice and the weight of the slice is determined by W. If we*

*assume that the soil is uniform and its density (γ) also, the weight of a slice of height h and width b can be written:*

$$W = \gamma bh \qquad \text{"}$$

**Enter on line 164-172:**

*The factor $r_u$ is a coefficient of pore pressure (interstitial pressure coefficient), which determines the rain infiltration factor on the slopes. As it is well known, the water that infiltrates the soil may produce a modification of the pore pressure, affecting its resistant capacity. This factor may vary from 0 (dry conditions) to 0.5 (saturated conditions). In the article of Spencer (Spencer, 1967), assuming a homogeneous pore-pressure distribution as proposed by Bishop and Morgenstern (1960), the mean pore-pressure on the base of the slice can be written like the equation 7."*

(1) Line 120, "precise" should be "accurate";

(2) we accept change

(1) Lines 125-126, "lower than 1, or stable if it is higher than 1" miss the (mathematical) possibility that Fs=1.

(2) The following change is introduced point (3), to have covered the Fs = 1.

(3) *"According to equations (4) and (5), the slope FOS (FS) can be considered unstable if its value is lower than 1, or stable if it is equal o higher than 1."*

(1) Moreover, the statement that Fs "tend to be higher than one" deserves an explanation in addition to Burbano et al. (2009), since it represents the whole point of the paper.

(2) As the sentence of the manuscript indicates:

*"It should be noted that, when applying the factor in the engineering and architecture fields,.............."*

In all the countries of the world in their technical regulations of application for civil engineering, architecture and geological engineering, include for the analysis of stability of slope the security coefficient , which include among other application regulations: The CTE (Technical code of The Spanish Building, Eurocode, Guide for Anchors and Stability of slopes of the Ministry of Development), among others.......................... ......

In addition to the aforementioned reference.

References:

CTE. Technical building Code, 2007. Basic Document Structural Safety Foundation DB-SE-C. Ministry of public works of Spain.

General Directorate of Roads of Spain, 2009. Foundation guide on road works. Madrid.

General Directorate of Roads of Spain, 2001. Guide for the design and execution of ground anchors in road works. Madrid.

Eurocode Building: https://eurocodes.jrc.ec.europa.eu/

We include that clarification in point (3)

(3) *"It should be noted that, when applying the factor in the engineering and architecture fields, the limiting value tends to be higher than 1, with common values being 1.2 or even up to 1.5 [Burbano et al., 2009], security coefficients that include The European technical regulations and, specifically, the technical regulations of Spanish application (table 2.1, of the DB-C of the CTE, or Technical Code of the Building) among others."*

(1) Lines 133-136: I do not understand the sentence from "However" to "exogenous factors of the slope". Moreover, as for the sentence "Fs>=1.3 can be considered stable by most standards", please se my detailed discussion below.

(2) As the sentence of the manuscript indicates:

No engineer or architect can think of calculating the stability of a slope, without taking into account a safety factor greater than unity, apart from the fact that it is prevented by the technical regulation of the states. A coefficient of 1.30 is not exaggerated; there are European regulations that raise it to 1.50.

In civil and geological engineering these coefficients what they do is foresee these exogenous agents, examples:

1. Construction on the side of a road, unforeseen overload ...........
2. Excessive infiltration not taken into account in other hillside calculus models ............
3. Construction of a building, subject to the slope of unforeseen overloads .................
4. Modification by man of the morphology of the slope, which makes it unstable ....................
5. Do not contemplate all the characteristics of the hillside or choose a bad model or calculation software ........

Among others.

(1) Lines 141-142: whay does the initial curve depend on the "data introduced"? Does the user specify the whole curve, or what? What is the dependence of the results upon such an initial, arbitrary choice?

(2) There is nothing arbitrary in our code, basically what the user of the code does is to introduce a cut point of the slope and a center point of the breaking curve and the code with those two points traces an initial circle of breakage, (yellow line) that corresponds to the initial break initial circle. Both initial data, are a common values to be introduced into a slope stability code.

The user also determines the number N of slices in which he wants to divide that slippery soil mass and the code programmed in matlab, by means of the Fsolve function he automatically calculates the Fs (safety factor) by the Spencer method of that initial curve.

The code does a non-linear calculation of the equations of the Spencer method. The originality of this code is that it automatically searches from that initial curve, the curve and the critical center with the function of Matlab Fmicon, with the restrictions imposed by the user and that come in the manuscript.

The user introduces the infiltration $r_u$ factor defined in the Spencer method, if $r_u = 0$, there is no infiltration and if we define a $r_u > 0$, we are introducing infiltration in the slope.

With these parameters entered in the code, this automatically calculates the safety factor of the slope and draws me the critical sliding curve of that topographic profile introduced. What we have shown in the case study, with the geotechnical tests carried out in situ, that the code works.

(1) Section 3 tries to describe in somewhat more detail the operation of the software, but this is also done in a confusing way, in my opinion. First, I find confusing to name a terrain stabilitt model as "terrain stability", but this might be my personal taste. Figure 2 shows a terrain profile, obtained from a DEM, and it is mentioned that the profile was splitted into 500 slices. How does the profile emerge from the DEM? Which DEM, with what resolution? Also, why 500 slices? Is this the number of DEM cells along the profile, or is it less, or more? If it is less, why is it so? If it is more, how do we interpolate the DEM and why? Is the use of a circular shape for the slip surface a limitation, which I believe it is, given that other engineering-like models use "trial" (and not "first", since there is no hierachy in the different "trial" surfaces) surfaces of any shape? For example, in the SSAP model, which has apparently an applicability domain
similar to the model presented in the manuscript and it is free as well, slip esurface can be of any shape, to my knowledge; I might be mistaking: https://www.ssap.eu. Fonts are way too small in any figure in the Manuscript.

(2) We believe that this section is largely answered with the previous ones. As for our model to define a circular form of sliding, we understand it is not a limitation as you say; because depending where this is the center of the critical circle of the slip curve, the shape of the surface will vary, our code being able to simulate both rotational and translational landslides.

With this methodology and the circular form you can simulate almost all existing forms of landslide. The SSAP model that comments, https://www.ssap.eu., Does not consider the hydrological factor of infiltration $r_u$ and the resolution of the equations by the Spencer method as our code.
It is different in its operation to ours and also only allows circular shapes; but as we have already mentioned, the breaking shapes of the landslide depends on where the center of the critical circle is.

When the center of the critical circle is further away from the slope, the shape of the break will be more like a translational landslide and

if the center of the critical circle is closer to the analyzed slope, we will be in a deep or rotational landslide.

(1) The introduction of wet conditions seems to be performed as a separate step, is it true? This is relevant, and really hard to understand. Why is the modification only considered "at the basis" of the terrain slice? This would account for a modified shear stress atthe bottom of the slice, but where is the contribution of water weigth along the whole slice? Is this approach rigorous, or is it an approximation? The curves in Fig. 3 are described to be different because of the "different data introduced": what kind of data did the Authors change? Introducing pore pressure effect is not "different data", it is a different physical mechanism, thus a different model model. The statement "after the outcome here, it can be stated that the rainwater infiltration factor is necessary to predict instabilities of the slope" contains two logical mistakes, in my opinion. Firstly,
for a "prediction" to be performed, one needs to have spatial and time dependence, or at least specify what is it that one is trying to predict, which is not the case here.
Secondly, to establish that infiltration is a "necessary" factor, it is not enough to show that results with and without inclusion of the pore pressure correction are different: one must show that the case with inclusion is closer to reality than the other case!
Lines 247-249: I do not understand the sentence "if this infiltration factor is small enough, taking into account the safety coefficients, the design may still be adequate, but there was a lack of critical information for calculating this parameter" is not only difficult to understand, it also poses severe doubts on how is it possible to develop/test a model in which rainfall infiltration is supposed to be one of the key ingredients, and then the test case is taken as one in which not enough data exist to apply the model itself??

(2) The operation of the proposed code has been explained in the previous section. The introduction of wet conditions as we have already mentioned is introduced with the $r_u$ factor, it is not true that it is a separate step; we in the example have made the comparison of introduction or not of the pore pressure, but it can be done directly. Precisely the originality of the code is that which is a factor integrated into the software itself. It is a rigorous approach and not approximate.

In case the analyzed slope is partially saturated as you indicate, the weight of the water in the slope is considered in the code when

entering the saturated density of the soil; instead of entering the dry density.

The difference of figure 3 with respect to the calculation of figure 2, is that a value of $r_u = 0.3$ has been introduced in the calculation, in figure 2 the infiltration of rainwater is not taken into account and in the Figure 3 the infiltration is considered. That is why there is a change in the safety factor and the shape of the critical curve. In figure 2 we have a stable slope Fs = 1.45 and in figure 3 we have an unstable slope when introducing the infiltration of rainwater into the slope (Fs = 0.95), in this comparison the interest of this is shown code in the stability analysis of slopes and landslides.

We do not agree that to predict must incorporate the time variable, since what is essential for landslide occurrence related to water infiltration, depends not so much on time, but on rainfall and also the lithological characteristics of the soil of the slope, as it can be among others the coefficient of permeability (K).

It is not true that the infiltration factor has not been introduced in the case study, a value of $r_u = 0.35$ has been introduced in the calculation and the code gave us a value of the slope safety factor of Fs = 0,95 (unstable), when in the dry state the code calculated a safety factor of Fs = 2,300 (stable).
The calculation of the safety factor in the STB2010 program; that lacks the analysis of infiltration in the calculation, offered a result of Fs = 2.063 (stable).
Using the STB2010 program, we would not have been able to previously detect the landslide of the case study of the manuscript, calculation that is not normally done in the stability calculations, with the calculation with our code we could have avoided the collapse of the building.

(1) Line 300: "dimensions" shold be "sizes", or something of the like. Information in lines 320 to 325 seems to be trivial enough not to be highlighted with a bulleted list. Moreover, the statement ".. after the event, accodring to the histogram" isvrather misterious, since I can't find any event in Fig. 7. In Section 4.2, Figure 7: when did the landslide considered in the paper occur, in the timeline? This is relevant information, is it not? The Authors refer to "level 2 and level 3": what are the levels the Authors refer to? They also refer to "infiltration calculations", when and how did they perform the

mentioned calculations? This is probably described in Section 4.3, but this comes out of the blue and I do not understand how the calculations were done, and why they were not described in the methodological Section, instead of the "input data" Section.

Section 4.3 is devoted to describe the results obtained using the proposed model with a real landslide scenario. This section is very confusing, again. First of all, the Authors compare theirs results with the results obtained from a different model/program; so far, so good - even if this should have been mentioned briefly in the Introduction and/or methology sections, since it is part of the reseach method. Then the refer to "previous calculations", about which the reader is not aware, and they discuss curves that are non existing in Figure 9 (yellow and red curves?). They pretend that the "curves are similar", without any attempt to quantify the extent to which they are similar. Of course they are similar indeed, since all of them are circles arcs, but that does not seem to me to be enough, as a comparison. The same goes for the comparison with the real landlside failure curve, which I do not understand if it was actually measured or not, or if it is measurable at all. Then, the Authors refer to measures in square meters of the "surface area that corresponded with the profile", which I do not understand. What does "correspond" mean? The software is supposed to provide a two-dimensional failure curve on a vertical plane, there is no corresponding surface area. Or, at least,

I don't see what it is, particularly I do not see what is the "real situation" the Authors refer to.

In the same Section, the Authors refer to a "very stable" slope as one with an Fs much larger than unity. I believe this is a conceptual mistake. A model in which slope stability is assessed with an Fs defined as the ratio of destabilizing forces to stabilizing ones, there is no such thing as "more stable". A slope, or a DEM cell, or a slice, is unstable if Fs<1, and stable otherwise. Different degrees of stability are not defined in the model, since no attempt whatsoever exist (in this and in similar models) to quantify the sensitivity of Fs results to the large number of parameters and assumtpions utilized to obtain the result, nor to give a measure of the uncertainty. We do not know if an enormous rainfall would change Fs by a tiny bit or by a large amount, nor what is going to happen if an earthquake comes about. In other words, values of Fs different from the exact value obtained from the calculations do not have different degrees of probability, thus different degrees of stability are undefined within such a model. At least, if no further analysis is performed. Lastly, in the same Section, six points are listed, which contains methodological remarks and no

results, As such, these do not belong in this Section, but to a previous one.

(2) The date of the landslide is shown in figure 7, it is precisely indicated in the histogram the month of February and March of 2010 in the plot, date of the landslide, after the accumulation of so many consecutive days of rainfall.

The levels come in table 1 of the manuscript, as a consequence of the geotechnical tests made by Geolen S.L.

A topographic profile of the slippery soil was made, using MDE and the results of the geotechnical tests of the Geolen laboratory (figure 6) and then compared with the curve shape calculated in our code (figure 10), giving a satisfactory result.

We refer to what has already been explained and clarified in the previous sections, for example when we talk about the safety coefficients, the higher the safety coefficient the more stable the slope.
As for the rain, we have already spoken in previous sections of the operation of the code and the possible uncertainties. We understand that this type of code is again confused with other stability programs based on probability.
As for the points of the section indicated, also by recommendation of other reviewers we will move to section 2 where appropriate.

---

## Author Comment (AC3) · 10 Jan 2019

We maintain that the manuscript is well within the aims of NHESS. We provide in the attached supplement, the constestations and contributions to the questions raised by you.

Thanks for your comments

regard greeting GENERAL COMMENTS

(1)The aim of this work/manuscript is the development of a software for single slope stability. A case study and a comparison with another software is presented. In my

opinion, the main originality of the paper is represented by the inclusion in the software of the infiltration effects, according to the lacking in other software slope stability based, but the used theory (Spencer's method) and the way to calculate the interstitial pressure on the slice base is well known. Therefore, the proposed model is not innovative and the authors should give more emphasis to the originality of the developed software, clarifying the advantages also in term of time simulation.

(2) We appreciate the interest in our work and thank you for your encouraging comments. Your suggestion of making more emphasis to the originality of the developed software is very important.

(3) failure curve and surface area, including the infiltration effects."

(1) I Suggest the authors to include a block diagram of the software in order to explain better their algorithm from the user definitions to outputs/results.

(2) It seems to us a very good suggestion for the understanding of the proposed algorithm. We have developed the block diagrams, according to the following figure:

(3)

Figure 11. Sequential TS algorithm (block diagrams). Numbers in parentheses refer to numbers in the text.

(1) Moreover, a sensitivity analysis of the parameters is missing: I suggest for example to add some plot, e.g., the safety factor varying the interstitial pressure coefficient ru, the center of failure curve, the number of slices, the density of soil, etc.

(2) A sensitivity analysis has not been considered here as we have introduced a 3.Terrain stability (TS) model behaviour tests.

(1) To me, in conclusion, the paper needs some improvements and major revisions should be required.

(2) We conduct all the improvements suggested by the reviewer and, we have made

changes to the SPECIFIC COMMENTS section as indicated by the reviewer, which substantially improve the final result of the manuscript.

Please also note the supplement to this comment:
https://www.nat-hazards-earth-syst-sci-discuss.net/nhess-2018-192/nhess-2018-192-AC3-supplement.pdf

**Supplement:**

**Comments by Editor:**

Dear Authors,

You - as the contact author - are requested to individually respond to all referee comments (RCs) by posting final author comments on behalf of all co-authors no later than 13 Jan 2019 (final response phase) at: https://editor.copernicus.org/nhess-2018-192/final-response.

**Comments by Anonymous Referee #3 (nhess-2018-192-RC3)**
[Answers in blue]

**GENERAL COMMENTS**

(1)The aim of this work/manuscript **is the development of a software for single slope stability**. A case study and a comparison with another software is presented. **In my opinion, the main originality of the paper is represented by the inclusion in the software of the infiltration effects, according to the lacking in other software slope stability based, but the used theory (Spencer's method)** and the way to calculate the interstitial pressure on the slice base is well known. Therefore, the proposed model is not innovative and the authors should give more emphasis to the originality of the developed software, clarifying the advantages also in term of time simulation.

(2) We appreciate the interest in our work and thank you for your encouraging comments. Your suggestion of making more emphasis to the originality of the developed software is very important.

(3) *failure curve and surface area, including the infiltration effects.*"

(1) I Suggest the authors to include a block diagram of the software in order to explain better their algorithm from the user definitions to outputs/results.

(2) It seems to us a very good suggestion for the understanding of the proposed algorithm. We have developed the block diagrams, according to the following figure:

(3)

[Figure]

*Figure 11. Sequential TS algorithm (block diagrams). Numbers in parentheses refer to numbers in the text.*

(1) Moreover, a sensitivity analysis of the parameters is missing: I suggest for example to add some plot, e.g., the safety factor varying the interstitial pressure coefficient ru, the center of failure curve, the number of slices, the density of soil, etc.

(2) A sensitivity analysis has not been considered here as we have introduced a 3.Terrain stability (TS) model behaviour tests.

(1) To me, in conclusion, the paper needs some improvements and major revisions should be required.

(2) We conduct all the improvements suggested by the reviewer and, we have made changes to the SPECIFIC COMMENTS section as indicated by the reviewer, which substantially improve the final result of the manuscript.

**SPECIFIC COMMENTS:**

(1) The **section 1 (Introduction)** must be expanded citing other works that develop/use stability model, e.g.:

(2) The comment seems right to us and we introduce the change. We introduce the proposed references and others from other authors and define slope stability models in the introduction to complete the study.

(3) *"Landslides, one of the natural disasters, have resulted into significant injury and loss to the human life and damaged property and infrastructure throughout the world (Crozier and Glade, 2005; Dai et al., 2002; Parise and Jibson, 2000; Varnes, 1996).*

*Normally, heavy rainfall, high relative relief and complex fragile geology with increased manmade activities, have resulted in increased landslide (Gutiérrez-Martin, 2015) It is essential to identify, evaluate and delineate landslide hazard prone areas for proper strategic planning and mitigation (Bisson et al., 2014). Therefore, to delineate landslide susceptible slopes over large areas, landslide hazard zonation (LHZ) techniques can be employed (Fall et al., 2006; Casagli et al., 2004; Guzzetti et al., 1999; Anbalagan, 1992).*

*Landslides are resulted because of intrinsic and external triggering factors. The intrinsic factors are mainly; geological factors, geometry of the slope (Wang and Niu, 2009; Ayalew et al., 2004; Anbalagan, 1992; Hoek and Bray, 1981).*

*The external factors which generally trigger landslides are rainfall (Dai and Lee, 2001; Collison et al., 2000; Anderson, 1985). Several LHZ techniques have been developed over the past and these can be broadly classified into three categories; expert evaluation, statistical methods and deterministic approaches (Canili et al., 2018; Zhang et al.; 2018; Lari et al., 2016; Raia et al., 2014; Rossi et al., 2013; Lu and Godt, 2008; Fall et al., 2006; Casagli et al., 2004; Crosta and Frattini, 2003; Inverson, 2000; Guzzetti et al., 1999; Leroi, 1997; Wu and Sidle, 1995). Within these models, we want to highlight the empirical models that are*

*based on rainfall thresholds (Matelloni et al., 2011; Gruzzetti et al., 2007; Aleotti, 2004; Wilson, 1997).*

*Each of these LHZ techniques has its own advantage and disadvantage owing to certain uncertainties on account of factors considered or methods by which factor data are derived (Carrara et al.,1995).*

*"Limit equilibrium types of analyses for assessing the stability of earth slopes have been in use in geotechnical engineering for many decades. The idea of discretizing a potential sliding mass into vertical slices was introduced in the 20th century. During the next few decades, Fellenius introduced the Ordinary method of slices (Fellenius, 1936) . In the mid1950s Janbu and Bishop developed advances in the method (Janbu, 1954; Bishop, 1955). The advent of electronic computers in the 1960's made it possible to more readily handle the iterative procedures inherent in the method, which led to mathematically more rigorous formulations such as those developed by Morgenstern and Price and by Spencer (Morgenstern and Price, 1965; Spencer, 1967)."*

**(1) There are plenty of free software (see for example TRIGRS model of USGS). You should cite them too and specify the differences with your model. Line 36-38.**

(2) To address this question the following text (including new references) has been added into the introduction section:

(3) *"Limit equilibrium types of analyses for assessing the stability of earth slopes have been in use in geotechnical engineering for las year. Currently, the vast majority of stability analyses using **this method of equilibrium limit** are performed with commercial software like SLIDE V5, SLOPE/W, Phase2, GEO-Slope, GALENA, GSTABL7, GEO5 and GeoStudio, entre otros [Mousavi, 2017; Acharya et al., 2016a; Acharya et al., 2016b; Jiao et al., 2013; Gonzalez de Vallejo et al., 2002). Other models of slope stability based on the theory of limit equilibrium are still being studied, as is the case of the SSAP model (Borselli, 2016), but in this case a General equilibrium method model is applied."*

*"There are other types of software based on the modeling of the probability of occurrence of shallow landslides LHZ, in more extensive areas using GIS technology and MDE, as is the case of deterministic software TRIGRS ,SINMAP, SHALSTAB, GEOtop/GEO-FS, R-Slope.stability among others (Tran et al., 2018;*

*Alvioli and Baum, 2016; Reid et al., 2015; Mergili et al., 2014a; Mergili et al., 2014b; Mergili et al., 2014c; Baum, 2008; Simoni et al., 2008; Rigon et al., 2006; Pack, 2001). They are widely used models for calculating the time and location of the occurrence of shallow landslides caused by rainfall at the territorial level; some even in three dimensions, in order to obtain a probabilistic interpretation of the factor of safety.*
*Currently other approaches / theoretical studies for landslide prediction are used (for triggering and / or propagation) (Matelloni et al., 2017; Martelloni and Bagnoli, 2014).*

*The idea of discretizing through this tool proposed (TS), the potential slip mass in the critical section of the slope, once we have detected through the HZD programs unstable areas, is one of the achievements of this model. This calculation tool is not limited to shallow landslides and debris flows, but allows analysis of deep and rotational landslides, which others do not allow. Using the infiltration factor of Spencer $r_u$, we introduce the hydrological variable by infiltration to the stability calculation of the slope."*

(1) The **section 2 (Terrain Stability model development)** needs some corrections:

The meaning of some parameters is missing in the text, e.g., in the equation 3 $R$ is the radius of the curvature and α is the angle of the slope referred to each slice (I suppose);

(2) The comment is correct and the change is introduced in lines 95-101 and 160-163.

(3) *"In this equation, Q is the resultant of the pair of forces between slices, and α is the angle of the resultant (Figure 1). From this, it can be stated that the sum of the moments of the forces between slices around the critical rotation centre is zero, conformed to equation 3:*

$$\sum [QR\,cos(\alpha - \theta) = 0] \qquad\qquad 3$$

*When the R is the radius of the curvature, α is the angle of the slope referred to each slice. This takes into account that the sliding surface is considered circular, so the radius of the curvature is constant."*

$$\text{``} \qquad u = r_u \gamma h \qquad\qquad\qquad 7$$

*In this expression, u is the pore pressure (permanent interstitial pressure) at the base of the slice, γ is the density of soil, h is the mean height of slice (if the height is not constant) and the weight of it affects the W evaluation."*

(1) In my opinion is not clear how the pore pressure is calculated by means of equation 7., i.e., how is the interstitial pressure coefficient ru calculated (according to heavy rainfall event)? Then, how does the equation 8 (Mohr-Coulomb law), for the calculus of u, come into play? In the article of Spencer (Spencer, 1967), assuming a homogeneous pore-pressure distribution as proposed by Bishop and Morgenstern (1960), the mean pore-pressure on the base of the slice can be written just like the equation 7 that is used for the calculation of the safety factor (substituting expression of u in equation 5). Please clarify the need of equation 8!

(2) The comment is correct and the change is introduced in lines 164-172. For the calculation of $r_u$, equation 8 is not necessary.

The pore pressure will be hydrostatic, defined by: $u = \gamma_w(h - h_w)$, $\gamma_w$ is the saturated density of soil, h and $h_w$ is the difference between saturated and dry height.

(3) *"The factor $r_u$ is a coefficient of pore pressure (interstitial pressure coefficient), which determines the rain infiltration factor on the slopes. As it is well known, the water that infiltrates the soil may produce a modification of the pore pressure, affecting its resistant capacity. This factor may vary from 0 (dry conditions) to 0.5 (saturated conditions). In the article of Spencer (Spencer, 1967), assuming a homogeneous pore-pressure distribution as proposed by Bishop and Morgenstern (1960), the mean pore-pressure on the base of the slice can be written like the equation 7.*

*This equation is used in our proposed algorithm for calculating the safety factor (substituting the expression of u in equation 5)."*

(1) The **section 3 (Terrain Stability (TS) model behaviour tests)**, in my opinion, should be renamed **Terrain Stability (TS) algorithm and tests** adding these points:
(2) The comment is correct and the change is introduced in lines 173.

(3) *"3. Terrain stability (TS) algorithm and tests"*

(1) I suggest including a block diagram of the software in order to explain in detail your algorithm from the user definitions to outputs/results.

(2) The comment is correct and the change is introduced in lines 173. The proposed diagram is introduced

(1) As sensitivity analysis of the parameters is missing, I suggest for example to add some plot, e.g., the safety factor varying the interstitial pressure coefficient $r_u$, the center of failure curve, the number of slices, the density of soil, etc.

(2) Due to the length of the article and its focus we do not consider this point.

**(1) Line 206:** It is not "centre", but center. Please, check the paper if other typos are present!

(2)The comment is correct and the change is introduced in line.

(3) *"The next step is to apply Spencer's method to the different breakage surfaces until the curve with the lowest $F_S$, is found, and that will be the critical surface susceptible to a circular slip. To determine the minimal Fs using this model, calculate the displacement of the lower cut point of the critical slip from slope, as well as the rotation center position of the critical failure curve."*

**Concerning the **section 4:**

**(1)Line 415:** I would not say "our innovative TS model", but "our original algorithm".

(2)The comment seems right to us and we introduce the change.

(3)*"As mentioned earlier, the STB 2010 model does not allow stability calculations to apply to rainfall infiltration on a hillside. Hence, it is not capable of predicting a hillside's instability in a critical rainfall scenario, which was critical in the slope analysed. The STB 2010 model found that the hillside studied had an Fs of FS = 2.063; that means it was a very stable slope. Consequently, our original algorithm TS model appears to be more efficient and accurate."*

**(1)Lines 421-444:** I would add this part in the section 3 where is requested the explanation of the algorithm (software).

(2)The comment seems right to us and we introduce the change.

---

## Author Response (AR1)

**Comments by Editor:**

Dear Authors,

You - as the contact author - are requested to individually respond to all referee comments (RCs) by posting final author comments on behalf of all co-authors no later than 26 Feb. 2019 (final response phase) at: https://editor.copernicus.org/nhess-2018-192/final-response.

**Comments by Anonymous Referee #1 (nhess-2018-192-RC1)**
[Answers in blue]

AC1-supplement, page 3, item 3 of 'SECTION 0: ABSTRACT': Consider rewording: "This model is especially useful for predicting the risk of landslides in scenarios of heavy unpredictable rainfall. We have called it (TS) Terrain Stability and programmed in MATLAB, which it allows us a simulation of the slope stability in a 2D spatial distribution. As originality in our algorithm a hydrological assumption has been incorporated in steadystate." to something like: "This model is especially useful for predicting the risk of landslides in scenarios of heavy unpredictable rainfall. The model, hereafter named 'Terrain Stability' or TS is a 2D model, programmed in MATLAB and includes a steady state hydrological term."

(2) The comment seems right to us and we introduce the change in the Abstract.

(3) *"This model is especially useful for predicting the risk of landslides in scenarios of heavy unpredictable rainfall. The model, hereafter named 'Terrain Stability' or TS is a 2D model, programmed in MATLAB and includes a steady state hydrological term."*

**Comments by Anonymous Referee #2 (nhess-2018-192-RC2)**
[Answers in blue]

AC1-supplement, page 5. Make sure to translate 'entre otros'.

(2) The comment seems right to us and we introduce the change
(3) *"among others"*

Authors, make sure to address the following comment provided by the 2nd reviewer: 'First, the Authors state that the proposed model "defines fairly well areas that intuitively appear to be susceptible to landslides and defined rigorously the failure curve". In this sentence, "fairly well" and "intuitively" are not good enough to assess the predicting performance of a quantitative model. Moreover, the "rigorous" definition of slip surfaces does not appear to be substantiated by the presented results, as I will explain at length in the following. Then, the expression "this model is probably the most powerful tool for determining slope stability", is again not substantiated by the presented results.' Provide additional information so the reader can better understand how the authors came to the conclusion that the model preforms 'fairly well', same with 'intuitively'. Additionally, by describing your model as 'the most' powerful tool assumes you did a thorough literature study and comparison with other models, so this paper would become more like a review paper, rather than a paper where a new model is presented. Maybe instead consider using: 'this model is a powerful tool for…'

(2) The comment seems right to us and we introduce the change several manuscript sections such as:

(3) In the Analytical results:

 *"We applied the TS model using topographic data obtained from the ArcGIS 10 software program. We did so to obtain the degree of stability of the sliding land based on the angle of internal friction, the cohesion, the density and the angle of the slope we analyzed. Figure 9 shows the analytical results from the real slope, by studying and analyzing the most unfavorable profile of the landslide studied. In addition we compared the results given by the developed TS model and the results given by STB 2010 model, using free surfaces in both cases. In our model the worst curve (shown in green) was calculated automatically from the initial curve (show in blue), resulting in FS = 2.300, in the dry state (Figure 12)."*

In the Conclusion:

*"……………….so we obtain a topographic map, a key element to obtain the topographic profile to be studied with our algorithm"*

*"……this model is a powerful tool for determining slope stability."*

---

## Editor Decision (ED1)

Dear Gutiérrez-Martín,

I would like to encourage you to address the below mentioned comments of the reviewers while incorporating your suggested changes. Especially comment 3 below deserves some attention. The 2nd reviewer commented on the wording in the conclusions section, in that some of the emphases are too strong on certain capabilities of the model without presenting quantitative evidence. I do agree with the reviewer that some rewording would be appropriate if no quantitative comparison can be presented. So please have a look at the comments below and address accordingly.

Best regards,
Albert.

1)
AC1-supplement, page 3, item 3 of 'SECTION 0: ABSTRACT':
Consider rewording: "This model is especially useful for predicting the risk of landslides in scenarios of heavy unpredictable rainfall. We have called it (TS) Terrain Stability and programmed in MATLAB, which it allows us a simulation of the slope stability in a 2D spatial distribution. As originality in our algorithm a hydrological assumption has been incorporated in steadystate." to something like:
"This model is especially useful for predicting the risk of landslides in scenarios of heavy unpredictable rainfall. The model, hereafter named 'Terrain Stability' or TS is a 2D model, programmed in MATLAB and includes a steady state hydrological term."

2)
AC1-supplement, page5. Make sure to translate 'entre otros'.

3)
Authors, make sure to address the following comment provided by the 2nd reviewer:
'First, the Authors state that the proposed model "defines **fairly well** areas that **intuitively** appear to be susceptible to landslides and defined **rigorously** the failure curve". In this sentence, "fairly well" and "intuitively" are not good enough to assess the predicting performance of a quantitative model. Moreover, the "rigorous" definition of slip surfaces does not appear to be substantiated by the presented results, as I will explain at length in the following. Then, the expression "this model is probably **the most powerful** tool for determining slope stability", is again not substantiated by the presented results.'

Provide additional information so the reader can better understand how the authors came to the conclusion that the model preforms 'fairly well', same with 'intuitively'.
Additionally, by describing your model as 'the most' powerful tool assumes you did a thorough literature study and comparison with other models, so this paper would become more like a review paper, rather than a paper where a new model is presented. Maybe instead consider using: 'this model is a powerful tool for…'

---

## Author Response (AR2)

**Comments by Editor:**

Thank you for submitting your modifications to your manuscript. I have read with pleasure, in detail the manuscript and appreciate your efforts. However, I have included below a few more suggestions to improve the manuscript.

**Comments by Editor (nhess-2018-192)**

[Answers in blue]

• Line 17-18. References shouldn't be included in an abstract. Please do include them but find better sections (like the introduction) to do so.

(2) The comment seems right to us and we introduce the change in the Introduction, lines 31-34.

(3) *"Landslides, one of the natural disasters, have resulted into significant injury and loss to human life and damaged property and infrastructure throughout the world (Varnes, 1996; Parise and Jibson, 2000; Dai et al., 2002; Guha-Sapir et al., 2004; Crozier and Glade, 2005; Kahn, 2005; Toya and Skidmore, 2007; Raghuvanshi et al., 2014; Girma et al., 2015)."*

• Line 32. Remove 'the' such that it reads: '…and loss to human life…'

(2) The comment seems right to us and remove 'the'.

(3) *"Landslides, one of the natural disasters, have resulted into significant injury and loss to human life........"*

• Line 40. Start new paragraph after 'Fall et al., 2006).', (instead of starting this paragraph at line 42-43).

(2) Ok, now it is:

(3) *".............. areas, landslide hazard zonation (LHZ) techniques can be employed (Anbalagan, 1992; Guzzetti et al., 1999; Casagli et al., 2004; Fall et al., 2006).*

*Landslides are resulted because of intrinsic and external triggering factors. The intrinsic factors are mainly; geological factors, geometry of the slope (Hoek and Bray, 1981; Ayalew et al., 2004; Wang and Niu, 2009). "*

• Line 48. Replace 'models' by 'categories' such that it reads: 'Within these categories, we want to highlight….'

(2) I accepted.

(3) *"Within these categories, we want to highlight the empirical ........"*

• Line 55. Replace 'next' by 'following' and move reference such that it reads: 'During the following few decades, Fellenius (1936) introduced…'.

(2) Ok, now it is:

(3*) "During the following few decades, Fellenius (1936) introduced ........"*

• Line 66. Write out 'MDE'.

(2) This is a misinterpretation from the Spanish, it should be DEM (Digital Elevation Model.

*(3) "........in more extensive areas using GIS technology and 'DEM' (Digital Elevation Model) ..."*

• Line 66. Replace 'software' by 'models like: '

(2) Ok, it is now, as follows:

(3*) "........models like: software TRIGRS, ......."*

• Line 70. Replace 'They are widely..' by 'These are widely…'

(2) Ok, this has been changed

(3*) "........These are widely......."*

• Line 74-77. This sentence is hard to read. Maybe rewrite sentence 'The idea of discretizing… of this model.', to something like: 'One of the achievements of the presented study is to discretize the potential slip mass

in the critical profile of the slope, once unstable areas have been detected through the HZD programs.'

(2) Ok, thank you.

(3*) "One of the achievements of the presented study is to discretize the potential slip mass in the critical profile of the slope, once unstable areas have been detected through the 'HZD' programs."*

• Line 76. Write out 'HZD'.

(2) Ok.

*(3) "…………the 'LHZ' (landslide hazard zonation) programs ………"*

• Line 77. Replace 'This calculation tool….' to something like: 'The TS calculation tool is…'

(2) Thank you.

(3*) "The TS calculation tool is…………"*

• Line 78. Replace '…which others do not allow.' To something like: '…which often other models do not accommodate for.'.

(2) thank you again.

(3*) "… which often other models do not accommodate for…………"*

• Line 79. I'm not sure that 'ru' is in '….Spencer ru we…'. Explain/correct.

(2) Ok. New wording and explanation introduced as follows:

(3*) "We use in our algorithm the hydrological variable 'r$_u$' of Spencer, to consider the infiltration of rainfall in the calculation of stability of the considered slope."*

• Line 82. Correct '… engineering for LAS year.'

(2) Ok, done.

*(3) "Limit equilibrium types of analyses for assessing the stability of earth slopes have been in use in geotechnical engineering for last years. "*

• Line 83. Add 'packages' so it reads: '….commercial software packages like….'

(2) ok, change done.

*(3) "………commercial software packages like……."*

• Line 86-87. 'are still being studied..'. I'm not sure what you mean by 'still being studied'. Are you studying them? Please better explain.

(2) Ok. New wording and explanation introduced.

*(3) "………Currently there are other slope stability models based on the theory of limit equilibrium that are still in analysis and testing, as is the case with the SSAP software package (Borselli, 2012)."*

• Line 88-89. Rephrase so it reads: 'Secondly, sometimes for commercial models, the introduction of parameters to perform calculations are not very interactive.'

(2) ok, change done.

*(3) "Secondly, sometimes for commercial models, the introductions of parameters to perform calculations are not very interactive."*

• Line 107. Rewrite so it reads to something lie: 'The above-mentioned software packages provide useful tools….'.

(2) ok, thanks.

*(3) "The above-mentioned software packages provide useful tools for determining………."*

• Line 173. Double check, not sure what the word 'FOS' means in 'the slope FOS (FS) can be …..'.

(2) ok, we have removed FOS. Now it is as follows:

(3*) "According to equations (4) and (5), the slope can be considered unstable if its value of the safety factor 'Fs' is lower than 1, ……….."*

• Line 246-247. I'm unsure what you mean by '…, the value of N is entered into the user code, plus divisions of the sliding mass, more accuracy but greater need for computer capacity'. Could amount of detailed information maybe left out? I'm not sure if it adds anything to better explain the study.

(2) New wording and explanation introduced.

(3*) "It must be taken into account that the mass susceptible to sliding must be divided into a sufficient number of slices. This value is entered into our code through the parameter 'N'. In the application example of our algorithm, the sliding mass was divided into N = 500 slices, this value of N is entered into the code by the user, who decides the value of that parameter. The greater the number of slices in which we divide the sliding mass, the calculation will be more accuracy. N = 500 slice, we consider it a balanced value for an optimal calculation, which relates two fundamental parameters (computer calculation capacity / capacity accuracy)."*

• Line 249. Reword the Figure 2 header. Maybe to something like: Figure 2. Idealized cross section of a slope. In this example, the center…'.

(2) ok, done.

(3*) "**Figure 2.** Idealized cross section of a slope. In this example, the center ……………."*

• Line 334. Change 'other model' to 'other models' (so plural).

(2) thanks.

(3*) "However, other models, ………"*

• Line 338. Replace 'This model' by 'The TS model…'

(2) ok, thank you.

(3*) "The TS model can ………."*

• Line 343. Replace 'Our model programme has another advantage…' to 'The TS model has an additional advantage…'

(2) Done.

(3*) "The TS model has an additional advantage: ….. "*

• Line 374-377. Rewrite such that it reads to something like: 'For this example, we used data of IGN, the Spanish National Geographic Institute(http://centrodedescargas.cnig.es/CentroDescargas), and downloaded bit map MTN25, thatis a 1: 25000 topographic map in ETRS89 coordinates and UTM projection.'

(2) Done.

(3*) "For this example, we used data of IGN, the Spanish National Geographic Institute (http://centrodedescargas.cnig.es/CentroDescargas), and downloaded bit map MTN25, that is a 1:25000 topographic map in ETRS 89 coordinates and UTM projection."*

• Remove line 380-382 'The map file…among others,', and rewrite the rest of the sentence such that it reads: 'Figure 8 shows the area of the case study.'

(2) Done.

(3*) "Figure 8 shows the area of the case study."*

• Line 383-384. Rewrite so it reads something like: 'From this map we obtained the topographic information to acquire all necessary profiles to study the landslide.

(2) Done.

(3*) "From this map we obtained the topographic information to acquire all necessary profiles to study the landslide."*

• Line 441. Rewrite 2nd sentence so it reads: 'Rainfall data has been provided by the Meteorological ….'

(2) Done.

(3) "Rainfall data has been provided by the Spanish Meteorological Agency (Station of Viñuela)."